# Double trouble: two retrotransposons triggered a cascade of invasions in *Drosophila* species within the last 50 years

Almorò Scarpa[1,2,6], Riccardo Pianezza [1,2,6], Hannah R. Gellert [3], Anna Haider[1], Bernard Y. Kim[3], Eric C. Lai [4], Robert Kofler [1] ✉ & Sarah Signor [5] ✉

Horizontal transfer of genetic material in eukaryotes has rarely been documented over short evolutionary timescales. Here, we show that two retrotransposons, *Shellder* and *Spoink*, invaded the genomes of multiple species of the *melanogaster* subgroup within the last 50 years. Through horizontal transfer, *Spoink* spread in *D. melanogaster* during the 1980s, while both *Shellder* and *Spoink* invaded *D. simulans* in the 1990s. Possibly following hybridization, *D. simulans* infected the island endemic species *D. mauritiana* (Mauritius) and *D. sechellia* (Seychelles) with both TEs after 1995. In the same approximate time-frame, *Shellder* also invaded *D. teissieri*, a species confined to sub-Saharan Africa. We find that the donors of *Shellder* and *Spoink* are likely American *Drosophila* species from the *willistoni, cardini*, and *repleta* groups. Thus, the described cascade of TE invasions could only become feasible after *D. melanogaster* and *D. simulans* extended their distributions into the Americas 200 years ago, likely aided by human activity. Our work reveals that cascades of TE invasions, likely initiated by human-mediated range expansions, could have an impact on the genomic and phenotypic evolution of geographically dispersed species. Within a few decades, TEs could invade many species, including island endemics, with distributions very distant from the donor of the TE.

Human activities have long influenced the evolutionary paths of various species, particularly through the widespread relocation of organisms across continents and ecosystems[1]. With climate change further disrupting natural habitats, many species are now expanding or shifting their ranges poleward in search of more favourable conditions[2–4]. This novel mobility applies to infectious agents as well - for example, crop pathogens have the opportunity for worldwide spread, which will bring them into contact with many new local species[5–7]. Habitat loss and increasing contact between humans and wildlife also creates novel opportunities for the spread of diseases[8–10].

Intragenomic invaders, such as transposable elements (TEs), may not be exempt from such dynamics - as species distributions change and humans transport organisms around the planet, they will come into contact with novel genomic invaders such as TEs.

TEs are stretches of selfish DNA that violate Mendelian inheritance by copying themselves to different genomic sites, thereby increasing their representation in gametes[11]. All eukaryotic genomes investigated so far harbour some proportion of TEs[12,13]. TEs may have a profound effect on the evolution of genomes, for example leading to structural rearrangements and the emergence of novel genes[14,15]. Insertions of

[1]Institut für Populationsgenetik, Vetmeduni Vienna, Vienna, Austria. [2]Vienna Graduate School of Population Genetics, Vetmeduni Vienna, Vienna, Austria. [3]Department of Biology, Stanford University, Stanford, California, USA. [4]Developmental Biology Program, Sloan-Kettering Institute, New York, New York, USA. [5]Biological Sciences, North Dakota State University, Fargo, USA. [6]These authors contributed equally: Almorò Scarpa, Riccardo Pianezza.
✉e-mail: rokofler@gmail.com; sarah.signor@ndsu.edu

TEs have also been responsible for spectacular examples of adaptation, such as the tail loss in hominoids, the industrial melanism of the peppered moth and insecticide resistance in fruit flies[16–18]. Nevertheless, it is likely that TEs are largely deleterious to hosts[19]. It is even feasible that the proliferation of TEs may drive populations or species to extinction[20–22]. Therefore, hosts have developed defence mechanisms which frequently involve small RNAs[23]. In *Drosophila*, small RNAs, termed piRNAs, mediate repression of TEs at the transcriptional as well as the post-transcriptional level[24–27]. These piRNAs are thought to be generated at distinct genomic loci, termed piRNA clusters[24,28]. Current theory predicts that a newly invading TE will be silenced when a copy of the TE inserts into such a piRNA cluster (i.e., the trap model)[28–33]. Most TEs are active in the germline, but some are active in the somatic tissue surrounding the germline[30,34–37]. They may generate virus-like particles in the soma to infect the germline. Likewise, there are specialised piRNA pathways in the germline and the soma to silence TEs[30]. piRNAs are produced from separate clusters for each of these pathways, and while the germline pathways relies on a dispersed set of clusters, the somatic pathway produces piRNAs primarily from a single piRNA cluster termed *flamenco*[30,34,38].

TEs which have been successfully suppressed in a given species will gradually accumulate mutations that may eventually render it immobile[39]. To escape death by gradual decay, TEs may be horizontally transferred to naive genomes that have not yet developed a defence against the TE[40–43]. The vector responsible for moving TEs across species boundaries is elusive, although viruses and parasitic mites may be involved[44–46]. Over the course of millions of years of evolution of insect and vertebrate lineages, many horizontal transfer events likely occurred[40,47]. However, very few recent cases of horizontal transfer have been documented. One particularly striking example concerns *D. melanogaster*, where at least eight TEs invaded within the last two centuries[41,48–52]. The best-documented case of a horizontal transfer is the *P-element*, transmitted from *D. willistoni* to *D. melanogaster* and spread in natural populations between 1950 and 1980[48,50,53]. Using strains of *D. melanogaster* sampled at different time points and historical museum specimens, it was also revealed that six more TEs invaded *D. melanogaster* prior to the *P-element*. Recently, we showed that the retrotransposon *Spoink* spread in *D. melanogaster* between 1983 and 1993[51].

Given the estimated number of known TE families in *D. melanogaster* (> 124) and the high number of TE invasions in the last 200 years, it is unlikely that this rate of invasions is the normal 'background' level of introduction. A more likely scenario is that human activity is indirectly responsible. Range expansions might bring previously isolated species into contact, thereby generating novel opportunities for horizontal transfer among these species. This is nicely illustrated by the two latest invaders of *D. melanogaster*, the *P-element* and *Spoink*, which were likely acquired from *D. willistoni* or a related species[51,53]. Since species of the *willistoni* group are endemic to Central and South America[54–56], horizontal transfer only became possible after *D. melanogaster* extended its distribution into the Americas ~200 years ago[57–59]. By facilitating range expansions, human activity indirectly influences the evolution of insect genomes.

Here, we investigated whether species closely related to *D. melanogaster* also experienced recent horizontal transfer of TEs. One such case, the invasion of *D. simulans* populations around 2010 by the *P-element* following horizontal transfer from *D. melanogaster*, has already been well documented[42,60]. *D. melanogaster* and *D. simulans* are both drosophilid species that have evolved different levels of human commensalism over the last 10,000 years and, as a consequence, have spread from solely African to cosmopolitan distributions[61–64]. Based on strains sampled during the last 70 years, we show that two LTR retrotransposons, *Shellder* and *Spoink*, invaded global *D. simulans* populations between 1995 and 2016, i.e., before the *P-element*. In contrast to *Spoink*, which is a germline TE, *Shellder* is a

somatic TE. In each analysed *D. simulans* strain we found exactly one *Shellder* insertion in the somatic cluster *flamenco*, providing support for the trap model for somatic TEs[31]. We show that following the invasion of *D. simulans*, *Shellder* and *Spoink* rapidly spread in populations of the island endemic species *D. mauritiana* and *D. sechellia*, likely after 1995. *Shellder* additionally spread to *D. teissieri* after 1995, a species inhabiting sub-Saharan Africa. Species endemic to Central or South America are the likely donors of *Shellder* and *Spoink*. We show that the cosmopolitan species *D. melanogaster* and *D. simulans* acted as hubs for transmitting *Shellder* and *Spoink*, thereby bridging the geographic gap between the American species and the species endemic to the Indian Ocean and Africa. Both the *Shellder* and *Spoink* invasions only became possible after the range expansion of *D. melanogaster* and *D. simulans* into the Americas 200 years ago. Our work reveals that cascades of TE invasions, initiated by human-mediated range expansion, may have a profound impact on the evolution of insect species, as many globally dispersed species can get infected by a TE within a short time.

## Results

### *Shellder* and *Spoink* invaded *D. simulans* populations between 1995 and 2015

Recently, a new retrotransposon, *Spoink*, was documented invading worldwide *D. melanogaster* populations between 1983 and 1993[51]. We asked whether *D. simulans* had also been recently invaded by previously undocumented TEs. *D. simulans* and *D. melanogaster* diverged 2–3 mya, and both became human commensals with a world-wide distribution[57,65–68]. Much as *D. melanogaster*, *D. simulans* also has the benefit of live strains collected over the last ~90 years. Therefore, we began by investigating long-read assemblies of old strains of *D. simulans* (*14021-0251.006*, collected in 1961 in Nueva, CA) and recently collected strains (*SZ232*, collected in 2012 in Zuma Beach, CA)[43,69,70]. The assembly *14021-0251.006* (1961) contained no full-length insertions of *Spoink*, whereas *SZ232* has multiple full-length copies (Supplementary Table S1). In addition to *Spoink*, a second TE annotated as *gypsy-29_DWil* (identified in *D. willistoni*; from the repeat library of ref. 71,72) also had substantially different copy numbers between the strains collected in 1961 and 2012 (Supplementary Table S1).

When extracting the sequence of the *gypsy-29_DWil* hits in *SZ232*, we realised they were identical (99.9%) to *Shellder*, a TE previously described by Ding et al.[73]. This retrotransposon was discovered because an insertion into an intron of *slowpoke* led to variation in the courtship song of a lab strain of *D. simulans*[73]. Since *Shellder* insertions were polymorphic, the authors suggested that *Shellder* was recently active in *D. simulans* and *D. mauritiana*[73]. Henceforth, we will continue with the name *Shellder*. *Shellder* and *Spoink* are both LTR retrotransposons, with a length of 6635nt and 5216nt respectively. *Spoink* has an LTR of 349nt and *Shellder* of 449nt. Both TEs encode for a *gag* and *pol* polyprotein, whereas *Shellder* additionally encodes for an *env* protein. Therefore, *Shellder* may be capable of generating virus-like particles that could infect other species or tissues (e.g., the germline), though this activity is, at present, largely hypothetical. A phylogeny based on the reverse-transcriptase domain shows that *Shellder* is a member of the *gypsy/gypsy* family (all members have an *env* protein) while *Spoink* belongs to the *gypsy/mdg3* family (members do not have *env*; Supplementary Fig. S1[51,74]). *Shellder* and *Spoink* insertions are both biased towards intergenic regions, though *Spoink* has a higher frequency of intronic insertions (30% vs 20%). Neither TE frequently inserts into other genic features (i.e., 5′ UTR, 3′ UTR, exon, promoter), whereas 42% of *P-element* insertions are found in promoters (Supplementary Fig. S2; based on *D. simulans*).

The presence of *Shellder* and *Spoink* in recently collected *D. simulans* strains and the lack of insertions in older strains suggest that these two TEs could have invaded *D. simulans* populations recently. To substantiate this hypothesis, we compiled a set of publicly available

short-read data sets comprising 88 *D. simulans* strains collected from different geographic regions during the last 70 years (for an overview of all analysed strains, see Supplementary Table S3;[60,69,75–79]). As a reference, we included the *P*-element in our analysis, which invaded *D. simulans* during the last 20 years[42,60]. We estimated the abundance of *Spoink*, *Shellder*, and *P*-element using DeviaTE, which infers TE copy numbers from short-read data[80]. Based on this method, we estimate that the *P*-element invaded *D. simulans* populations between 2000 and 2015 (Fig. 1A, Supplementary Dataset S1), which is in agreement with previous works[42,60]. This suggests that our approach, based on 88 sampled *D. simulans* lines from different regions, allows us to capture the approximate timing of TE invasions. We further estimate that *Shellder* and *Spoink* spread in *D. simulans* populations between 1995 and 2015, predating the invasion of the *P*-element (Fig. 1A). The appearance of *Shellder* and *Spoink* in one strain collected 1975 in Kenya (*Ken75*, SRR22548178) is possibly due to contamination (although it is possible that the invasion of both TEs already began in 1975). The coverage of *Shellder* and *Spoink* was uniformly elevated in strains having these two TEs (copy number estimate >1), whereas few reads mapped to these TEs in strains collected prior to the invasion (Supplementary Fig. S3). This suggests that the elevated copy numbers of these two TEs are not due to a mapping artefact, where for example a high number of reads aligning to a small region of the TE drive unusually high copy number estimates. Furthermore, long-read assemblies of *D. simulans* strains collected before 1994 do not possess *Shellder* and *Spoink* insertions, whereas strains collected between 1994 and the present do (Supplementary Table S4). When investigating the geographic spread, we found that both TEs were absent in strains sampled prior to 1994 (ignoring the putatively contaminated strain *Ken75*) but were latter found in many strains collected from diverse geographic regions. This suggests that *Shellder* and *Spoink* rapidly spread in worldwide *D. simulans* populations (Fig. 1B). However, due to the rapidity of this spread, we cannot trace the geographic route of these invasions. Interestingly, we found that *Shellder* and *Spoink* insertions were always co-occurring in *D. simulans* strains, i.e., both TEs were either jointly present or absent in a given *D. simulans* strain (Supplementary Fig. S4). This raises the possibility that *Shellder* and *Spoink* are co-transposing as a unit (e.g., if one TE is nested in the other). However, an analysis of the insertion sites in a long-read assembly (*SZ129*[43]) shows that the insertion sites of these two TEs are at entirely different genomic locations, suggesting that these two TEs are independently transposing (Supplementary Fig. S5).

*Spoink* does not encode an *env* protein necessary for cell-to-cell transmission and we therefore expect that *Spoink* is active directly in the germline. In agreement with this, we previously found ping-pong signatures for *Spoink* in strains sampled from natural *D. melanogaster* populations, suggesting that *Spoink* is silenced by the germline piRNA pathway[51]. We also found that the number of *Spoink* insertions in canonical dual-strand piRNA clusters of the germline is very heterogeneous among strains. Some strains have several insertions into *42AB*, while others have no *Spoink* insertion in canonical clusters[51]. The invasion of *Shellder*, a TE that is likely active in the soma (*Shellder* has an *env* gene), provides us with the opportunity to study the establishment of the host defence against somatic TEs. Ovarian small RNA from *D. simulans* strains collected in 2013 in Chantemesle (France)[81] showed piRNAs along the sequence of *Shellder* as well as *Spoink* (Supplementary Fig. S5). piRNAs mapping to *Shellder* were limited to the antisense strand, whereas *Spoink* piRNA was found on both strands (Supplementary Fig. S5). In addition, a ping-pong signature, i.e., the hallmark of an active germline piRNA pathway, can be found for *Spoink* but not for *Shellder* (as expected for a somatic TE; Supplementary Fig. S5).

If *Shellder* is a somatic TE we would expect it to insert into *flamenco*, as this is the primary piRNA cluster of the somatic piRNA pathway[30,38]. In *D. simulans*, *flamenco* has duplicated, such that there

are two paralogous copies[43]. In strains of *D. simulans* which carry *Shellder*, we found exactly one insertion into *flamenco* or its duplicate, whereas no *Spoink* insertions were present in either copy of *flamenco* (Supplementary Fig. S6 and supplementary Table S4). Importantly, the *Shellder* insertions in *flamenco* are at 8 different sites, suggesting that each *flamenco* insertion is due to an independent insertion event (Supplementary Fig. S6). Under the trap model, a single insertion of *Shellder* into *flamenco* is expected (i.e., a TE is active until one copy randomly inserts into a piRNA cluster[31,82]). Hence, *Shellder* meets predictions under the trap model for the somatic piRNA pathway in *D. simulans*. We did not investigate *Spoink* insertions in germline clusters in *D. simulans*, since germline clusters are rapidly evolving in *Drosophila*[83–85], making reliable identification of germline clusters in *D. simulans* strains challenging (the location of the somatic cluster *flamenco* can be identified in *D. simulans*[43,86]).

In summary, we show that the two LTR retrotransposons, *Shellder* and *Spoink*, spread in worldwide *D. simulans* populations between 1995 and 2015, prior to the invasion of the *P*-element. *Shellder* is likely active in the somatic tissue of the ovary, while *Spoink* is active in the germline. Furthermore, *Shellder* and *Spoink* seem to be controlled by the piRNA pathway in *D. simulans* (Supplementary Fig. S5). For *Shellder*, many independent insertions in *flamenco* were observed, consistent with the predictions of the trap model.

### *Shellder* was not found in *D. melanogaster*

We previously found that *Spoink* spread in *D. melanogaster* populations between 1983 and 1993[51]. Here, we provide evidence that both *Shellder* and *Spoink* invaded *D. simulans* between 1995 and 2015, leading us to consider whether *Shellder* has also invaded *D. melanogaster*. To investigate this, we estimated the abundance of *Shellder* in a set of 183 *D. melanogaster* strains sampled during the last 200 years. As a reference, we also included *Spoink* and the *P*-element. Consistent with previous work, our data show that the *P*-element spread in *D. melanogaster* populations between 1960 and 1980, whereas *Spoink* spread between 1983 and 1993 (Supplementary Fig. S7, refs. 48,50,51). However, *Shellder* is absent in all analysed strains of *D. melanogaster*, both in 179 short-read data and in 49 long-read assemblies (Supplementary Table S2 and Supplementary Fig. S8 refs. 87–90), including the most recent strain collected in 2018 in Kiev, Ukraine. Thus, we concluded that, as of 2018, *Shellder* has not invaded *D. melanogaster* populations.

### *Shellder* and *Spoink* invaded *D. mauritiana* and *D. sechellia*

*D. simulans* is closely related to both *D. mauritiana* and *D. sechellia*, having diverged from them around 250,000 years ago[71,91,92]. *D. simulans* hybridises with *D. mauritiana* and *D. sechellia*[93–95], which may facilitate the spread of *Shellder* and *Spoink* across species barriers. To investigate this, we obtained short-read data for 12 *D. mauritiana* strains collected during the last 50 years (Supplementary Table S6)[91,96–98]. We estimated the abundance of *Shellder* and *Spoink* insertions in these strains using DeviaTE, again utilising the *P*-element as a reference. Consistent with previous works, the *P*-element was absent from all investigated *D. mauritiana* strains (Fig. 2[42] and Supplementary Dataset S2). In contrast, *Shellder* and *Spoink* were absent in 3 strains collected before 1980 but present in all 9 strains collected in 2006 (Fig. 2A). This finding is in agreement with the hypothesis of[73] that *Shellder* is recently active in *D. mauritiana*. *Shellder* and *Spoink* also co-occur in *D. mauritiana*, where both TEs are either jointly present or absent in any strain (Supplementary Fig. S9). To substantiate the finding that both *Shellder* and *Spoink* invaded *D. mauritiana*, we generated long-read assemblies of four recently collected *D. mauritiana* strains (*R61*, *R39*, *R32*, *R31*; collected in 2006). We found insertions of *Shellder* and *Spoink* in both strains but not in the reference strain *14021-0241-01* (David 105, likely collected in 1981[99,100]) nor in the strain *w*[12] which was collected after 1970 (Supplementary Fig. S10)[100,101]. Again, *Shellder* and *Spoink* were co-occurring in the long-read

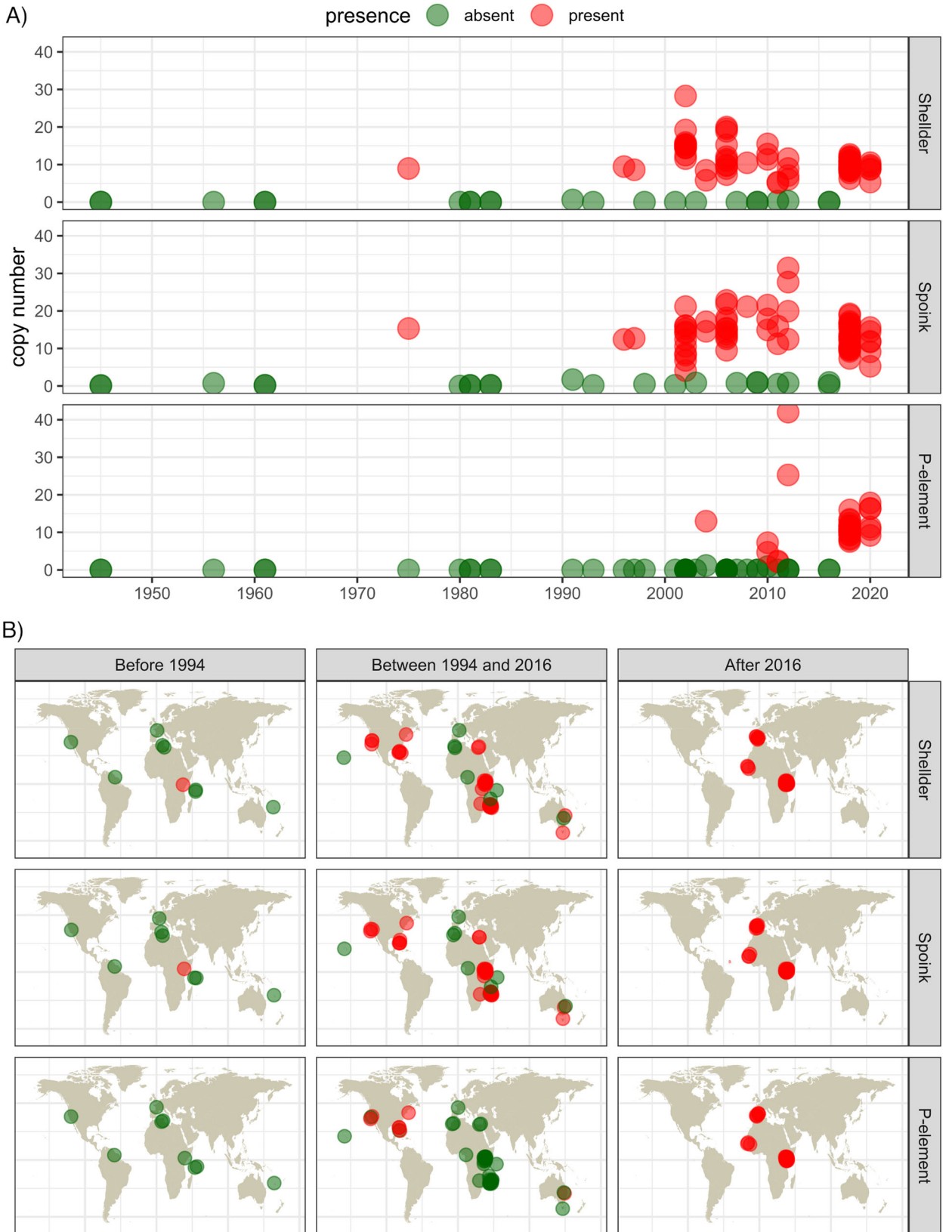

**Fig. 1 | *Shellder* and *Spoink* spread in *D. simulans* populations between 1995 and 2015. A** Abundance of *Shellder, Spoink,* and *P-element* in different *D. simulans* strains collected during the last 70 years. **B** Geographic spread of *Shellder, Spoink,* and *P-element* in worldwide *D. simulans* populations during the last centuries. Strains having a given TE are shown in red while strains without the TE are green.

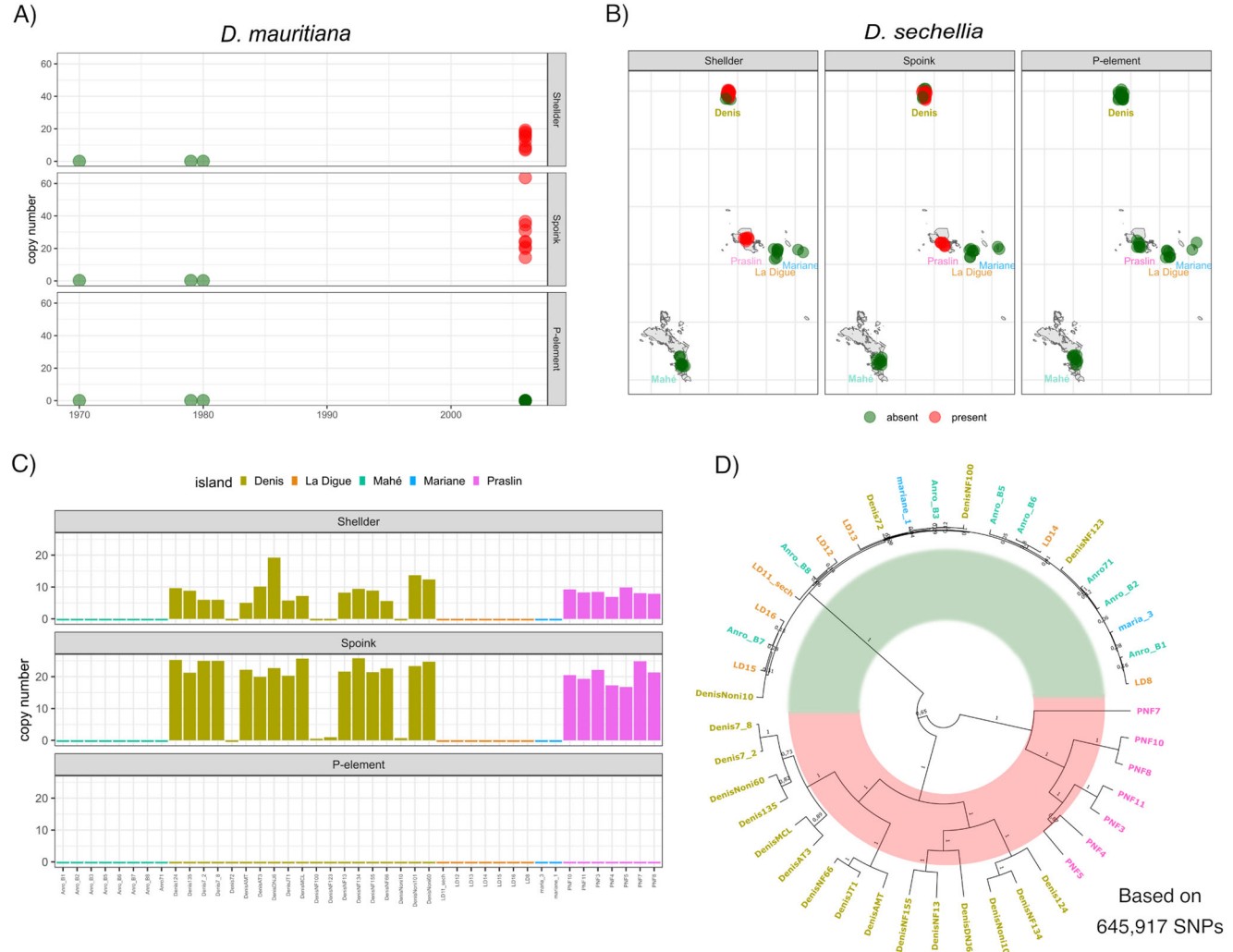

**Fig. 2 | *Shellder* and *Spoink* recently invaded *D. mauritiana* and *D. sechellia*.**
**A** Abundance of *Spoink*, *Shellder* and the *P-element* in *D. mauritiana* strains collected between 1970 and 2006. **B** Presence (red) or absence (green) of the TEs in *D. sechellia* strains collected in different islands of the Seychelles. **C** Copy number estimates of *Shellder*, *Spoink*, and the *P-element* for all *D. sechellia* strains. Colours refer to different islands. **D** Relatedness among the different *D. sechellia* strains based on > 600,000 autosomal SNPs. Note that strains with (red bar) and without (green bar) *Shellder* and *Spoink* form two distinct clades.

assemblies of strains *R31*, *R32*, *R39*, and *R61*. (Supplementary Fig. S10). Insertions of the two TEs are at entirely different sites in *D. mauritiana*, thus the two TEs are not transposing as a unit (Supplementary Fig. S7).

Next, we investigated the abundance of *Shellder* and *Spoink* in *D. sechellia*. While data for different time points are not available, short-read data are available for 43 *D. sechellia* strains collected in 2012 from five different islands in the Seychelles archipelago (Denis, Praslin, La Digue, Mariane, Mahé; Supplementary Table S8[92,93]). As expected, the *P-element* was absent in all *D. sechellia* strains[42]. Interestingly, *Shellder* and *Spoink* were confined to a subset of the islands of the Seychelles archipelago. *Shellder* and *Spoink* could be found in strains collected from Denis and Praslin, but were absent from Mariane, Mahé and La Digue (Fig. 2B). In Praslin, *Shellder* and *Spoink* were present in all strains (7/7), while in Denis they could only be found in 15 out of 19 strains (Fig. 2C). Therefore, *Shellder* and *Spoink* were also co-occurring in *D. sechellia*. There is just one long-read assembly available for *D. sechellia* strain *sech25*, which is derived from a strain collected around 1981 and inbred for 9 generations (Corbin Jones, pers. comm.[102,103]). In this assembly we found *Spoink* insertions but no *Shellder* insertions, raising the possibility that *Spoink* invaded *D. sechellia* slightly before *Shellder*. Alternatively, it is feasible that *Shellder*, but not *Spoink* was lost during the process of inbreeding.

The geographical structure of *Shellder* and *Spoink* invasions in *D. sechellia* suggests population structure between the islands. We investigated the relatedness among *D. sechellia* from the five islands by generating a phylogenetic tree based on > 600, 000 autosomal SNPs. Strikingly, strains with and without the two TEs form two distinct clades (Fig. 2D). Strains without *Shellder* and *Spoink* do not group by island and are not genetically differentiated. In contrast, the strains with *Shellder* and *Spoink* cluster based on the island (Denis and Praslin) with much deeper branches in the phylogenetic tree. We confirmed this result with a PCA based on autosomal SNPs (Supplementary Fig. S11).

To summarise, we found that both *Shellder* and *Spoink* have invaded *D. mauritiana* during the last 50 years. Based on a patchy geographic distribution in the Seychelles, where some island populations have the two TEs while others do not, we infer that *Shellder* and *Spoink* also invaded *D. sechellia* recently. The pronounced population structure observed in *D. sechellia* may be linked to barriers affecting the spread of these two TEs among the islands.

### *Shellder*, but not *Spoink*, was found in *D. teissieri*
As both *Shellder* and *Spoink* have spread to multiple species within a short period, we investigated whether these two TEs also invaded

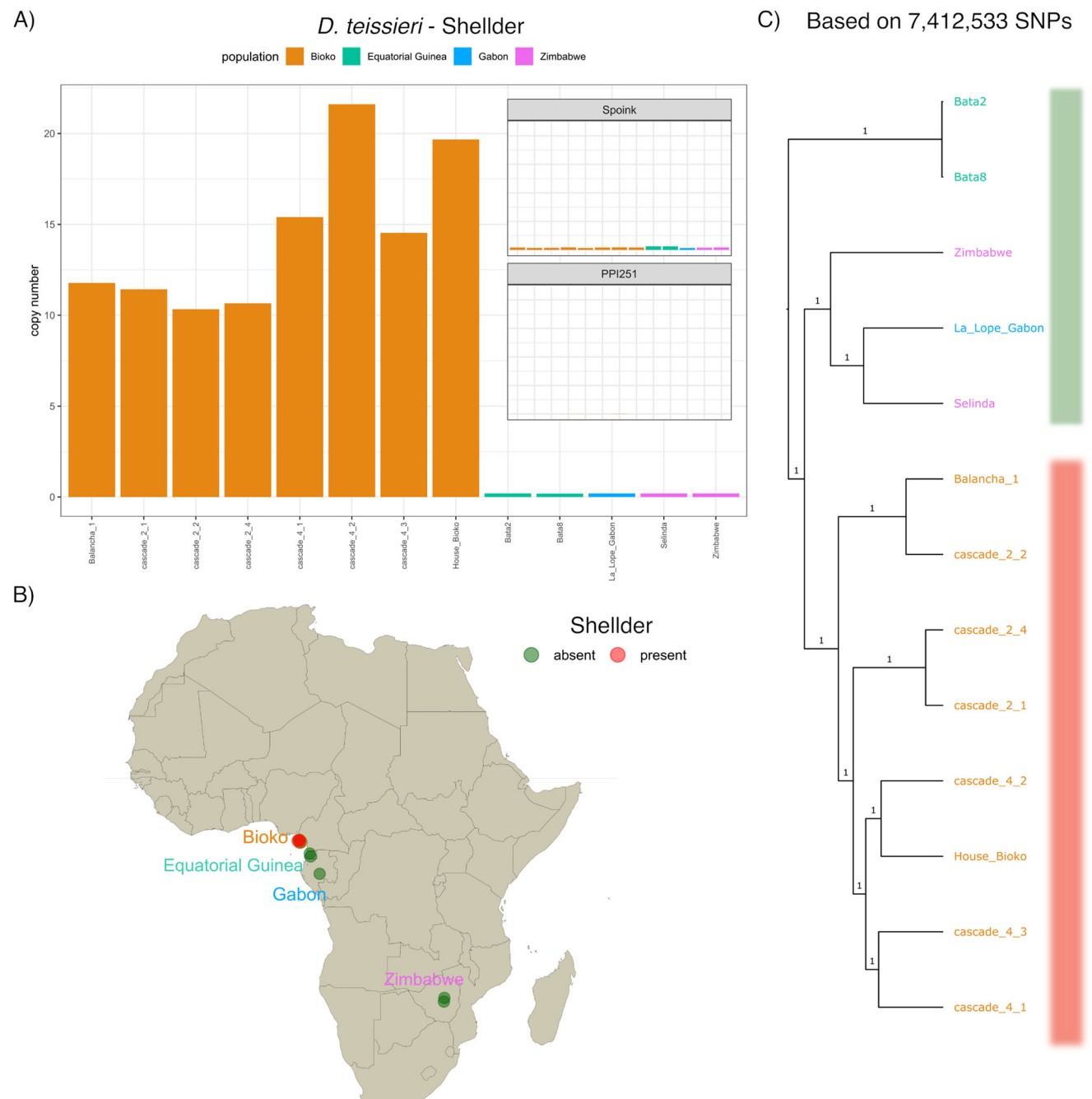

**Fig. 3 | *Shellder*, but not *Spoink*, invaded *D. teissieri* population. A** Abundance of *Shellder* in different *D. teissieri* strains sampled at different geographic origins (colours). *Spoink* and the *P-element* are shown in the inlays. **B** Presence (red) or absence (green) of *Shellder* in *D. teissieri* strains collected from different geographic locations. **C** Relatedness among the different *D. teissieri* strains based on >7*mil* autosomal SNPs.

other species within the *melanogaster* subgroup. *D. teissieri* which is endemic to sub-Saharan Africa and ranges from Equatorial Guinea to Zimbabwe[67], diverged from *D. melanogaster* approximately 12 million years ago[104]. We first analysed publicly available short-read data from 13 *D. teissieri* strains (ref. [105]; Supplementary Table S9). These strains were sampled from several geographic regions in Africa (Fig. 3A, B, Supplementary dataset S3). However, there is some uncertainty regarding the exact sampling date of the strains. While it is clear that the strains from Bioko were sampled around 2013, it is likely that the strains from continental Africa were sampled earlier. We found that *D. teissieri* populations from Bioko island harbour *Shellder* insertions, but those from continental Africa do not (Fig. 3A). Among the two available long-read assemblies for *D. teissieri*, strain *273.3* has *Shellder* insertions

while *ct02* does not have them (Supplementary Fig. S12). Both strains were collected in Cameroon[106] (date unknown), suggesting that *Shellder* may have spread to mainland Africa. This geographically (and possibly temporal) patchy distribution suggests that *Shellder* invaded *D. teissieri* very recently. In addition, *Spoink* and the *P-element* are absent in all investigated *D. teissieri* strains (Fig. 3A), and both long-read assemblies of *D. teissieri* lack any *Spoink* and *P-element* insertions (Supplementary Fig. S12). The absence of the *P-element* in *D. teissieri* is consistent with previous work[107]. Thus, *D. teissieri* is the first newly invaded species where we find *Shellder* but not *Spoink* (the reverse is true for *D. melanogaster*).

We investigated whether the geographically patchy distribution of *Shellder* might be linked to population structure in *D. teissieri*, as was

observed in *D. sechellia*. To test this, we inferred the relationship among *D. teissieri* strains based on more than 7 million autosomal SNPs. Indeed, strains from Bioko island and mainland Africa form separate clades (Fig. 3A). This suggests that the population structure in *D. teissieri* reflects the presence or absence of *Shellder*, potentially due to geographic barriers limiting the gene flow between island and continental populations. Given the uncertainty in sampling dates, it is also possible that the mainland African populations were sampled before *Shellder* had spread to *D. teissieri*. In summary, the geographically patchy distribution suggests that *Shellder,* but not *Spoink,* recently invaded *D. teissieri*. The presence/absence pattern of *Shellder* insertions in geographically distinct populations is linked to their population structure.

### *Shellder* and *Spoink* were not found in other species of the *melanogaster* subgroup

*D. yakuba* is closely related to *D. teissieri,* and the habitats of the two species are largely overlapping[67,104]. *D. teissieri* and *D. yakuba* produce hybrids in the laboratory[108], and these two species form a narrow hybrid zone in Bioko island[109], i.e., the island where *Shellder* has been found in *D. teissieri*. Given that hybridisation may facilitate the spread of the TE among species, we were interested in whether *D. yakuba* is also invaded by *Shellder*, and possibly by *Spoink*. We did not find *Shellder* and *Spoink* in short-read data of 38 *D. yakuba* strains sampled between 2009 and 2013 from different geographic regions including Bioko island, São Tomé, mainland Africa and the island of Mayotte (Supplementary Tables S10, S12). Similarly, both *Shellder* and *Spoink* were not found in other species of the *melanogaster* subgroup: we did not find these two TEs in short-read data of 17 *D. santomea* strains, 4 *D. erecta* strain and one *D. orena* strain (Supplementary Tables S11, S12;[93,110,111]). *Shellder* and *Spoink* were also absent in the long-read assembly of *D. yakuba* and *D. erecta* (Supplementary Fig. S12). The *P-element* is also absent from all four species (Supplementary Table S12 and Supplementary Fig. S12). For *D. yakuba,* the absence of the *P-element* is in agreement with previous works (*D. santomea, D. erecta, D. orena* were not investigated[53,107]). The most likely conclusion is that *Shellder* and *Spoink* have not yet invaded *D. yakuba, D. santomea, D. erecta*, and *D. orena*, though additional sampling is needed.

### Origin of *Shellder* and *Spoink*

The invasions of *Shellder* and *Spoink* in several species of the *melanogaster* subgroup were likely triggered by horizontal transfer. Based on the long-read assemblies of 101 *Drosophila* species, we previously suggested that the *Spoink* invasion in *D. melanogaster* was triggered by a horizontal transfer from a species of the *willistoni* group[51,106]. Since then, more then 100 novel assemblies of *Drosophila* species have been released[112]. We examined the presence of both *Shellder* and *Spoink* in these publicly available assemblies. We also included our 4 novel assemblies of *D. mauritiana* strains (*R61, R39, R32, R31*), two assemblies of *D. simulans* (*SZ129, SZ232*[43,84]) and 8 assemblies of *D. melanogaster* (*Es-Ten, Se-Sto, RAL91, RAL176, Pi2, RAL737, RAL732, Iso1*[84,90,113]; for an overview of all 266 investigated drosophilids assemblies see Supplementary Dataset S5). In addition, we analysed the reference genomes of 1226 arthropod species (see Supplementary Dataset S6). We used RepeatMasker to identify *Shellder, Spoink* and *P-element* insertions in these genomes.

Aside from the recently invaded species of the *melanogaster* subgroup (*D. melanogaster, D. mauritiana, D. simulans, D. sechellia, D. teissieri*), we found insertions with high similarity (based on Smith-Waterman score; see M&M) to *Spoink* and the *P-element* in species of the *willistoni* group (Fig. 4A; turquoise shades; Supplementary Fig. S12; for an overview of the insertions with the highest similarity, including the genomic location, see Supplementary Dataset S7). No insertions of *Spoink* and the *P-element* were found in any of the analysed arthropod species (Supplementary Fig. S13). This is in agreement with previous

works suggesting that a species of the *willistoni* group is a donor of *Spoink* and the *P-element*[41,51,53]. On the other hand, insertions with high similarity (based on Smith-Waterman score) to *Shellder* were found in species of the *cardini* and the *repleta* groups (Fig. 4A and Supplementary Dataset S4) pink and grey shades) and to a lesser extent in species of the *willistoni* and *saltans* groups (Fig. 4A); Supplementary Fig. S12). Thus, both TEs exhibit a patchy distribution among the investigated species, supporting the hypothesis of horizontal transfer (Fig. 4A); refs. 44,114,115). Interestingly, we also found *Shellder* insertions in *Anastrepha ludens* and *Anastrepha obliqua*, two important pest species of the *Tephritidae* family, that are native to Central America (Supplementary Fig. S13 ref. 116). The *Shellder* invasion in these *Anastrepha* species was likely triggered by horizontal transfer from a species of the *saltans* group, crossing an evolutionary distance of approximately 60 million years (Supplementary Fig. S13; ref. 117).

To investigate the source of the horizontal transfers more carefully, we generated phylogenetic trees of full-length insertions of *Shellder* and *Spoink*. These phylogenetic trees reveal a high level of discordance between the trees based on host genes and those based on the TEs. This discordance supports the hypothesis of horizontal transfer for both *Shellder* and *Spoink* (Supplementary Fig. S14; refs. 44,114,115). Moreover, insertions of both *Shellder* and *Spoink* in *D. melanogaster, D. simulans, D. mauritiana, D. sechellia*, and *D. teissieri* have short branches, consistent with a recent invasion (Fig. 4B, C). *Spoink* insertions of the *melanogaster* subgroup are nested within the *willistoni* subgroup (i.e., the branches from the *melanogaster* group are contained within species of the *willistoni* group; Fig. 4C), in agreement with our previous work, suggesting that a horizontal transfer from a species of the *willistoni* group triggered the recent *Spoink* invasions[51]. For example, the sequence identity between the consensus sequence of *Spoink* and insertions in *D. willistoni* is very high (99.59% over 4662 bp). On the other hand, the *Shellder* insertions of the *melanogaster* subgroup are nested within species of the *cardini* and *repleta* groups (Fig. 4B), suggesting that a horizontal transfer from one of these species triggered the recent *Shellder* invasions. The sequence identity between the consensus sequence of *Shellder* and insertions in these species is very high (e.g., *D. cardini* 99.31% over 6640 bp and *D. anceps* 99.29% over 6639 bp).

Finally, we investigated the divergence of the TEs and the host genes in pairs of species that were likely involved in the horizontal transfer of *Spoink* and *Shellder*. If the divergence of a TE is lower than the divergence of the host genes, this is usually considered as evidence for horizontal transfer of a TE[44,114,115]. For both *Spoink* and *Shellder*, the divergence between the species is much lower than the divergence of the host genes (*Spoink*: *D. melanogaster* and *D. willistoni, D. simulans* and *D. willistoni*; *Shellder*: *D. simulans* and *D. anceps*) strongly supporting horizontal transfer of these TEs. Although *Shellder* and *Spoink* are co-occurring in strains from several species (e.g., *D. mauritiana, D. simulans, D. sechellia*), our data suggest that the invasions of these two TEs were triggered by horizontal transfer from two distinct species groups. Nevertheless, species of the *cardini, willistoni* and *repleta* groups are endemic to Central and South America[54–56]. Therefore, both the *Spoink* and the *Shellder* invasions were triggered by horizontal transfer from a South American species.

To summarise, the *Spoink* invasion was likely triggered by a horizontal transfer from the *willistoni* group, whereas the *Shellder* invasion was triggered by horizontal transfer from the *cardini* or *repleta* group. *Shellder* also spread from a species in the *saltans* group to two *Anastrepha* species, bridging an evolutionary distance of about 60 million years.

### Reconstructing the cascade of TE invasions

We aimed to reconstruct the order of the horizontal transfer events that led to the invasion of *Shellder* and *Spoink* in several species of the *melanogaster* subgroup (Fig. 5A). Unfortunately, the sequence of

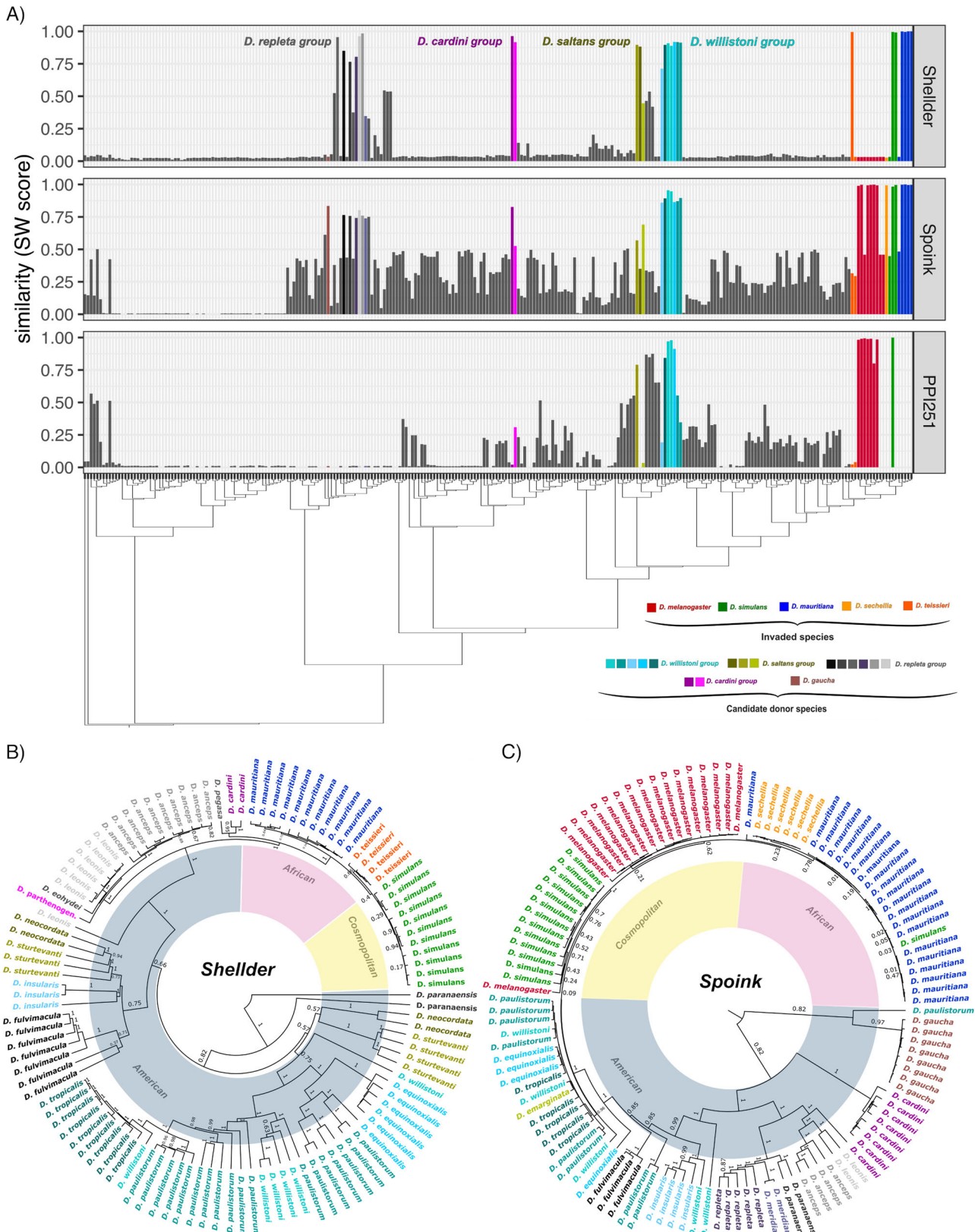

**Fig. 4 | The *Shellder* invasions were likely triggered by a horizontal transfer from a species of the *cardini* or *repleta* groups, whereas the *Spoink* invasions originated from a species of the *willistoni* group. A** Similarity of the consensus sequence of *Shellder*, *Spoink* and the *P-element* with TE insertions in 266 long-read assemblies from 243 drosophilids species. The bar plots show, for each assembly, the similarity (based on the Smith-Waterman alignment score; see M&M) between the given TE and the best match in an assembly. The species were arranged by relatedness based on a tree inferred from BUSCO genes. **B**, **C** Bayesian tree of *Shellder* and *Spoink* insertions in species having at least one complete insertion (see M&M). Multiple entries for a single species represent different insertions of a TE family in a single assembly.

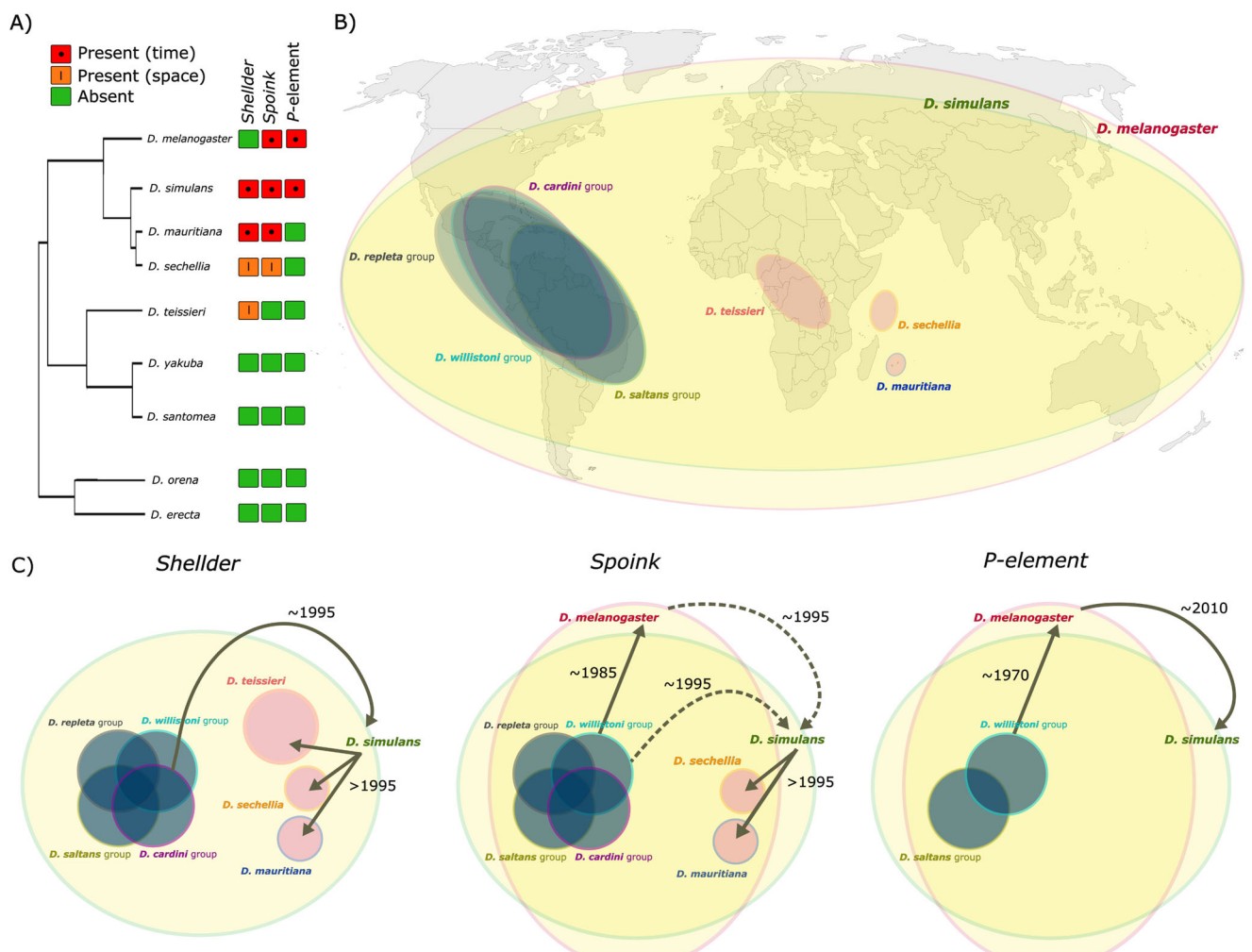

**Fig. 5 | Reconstructing the cascade of transposon invasions in the *melanogaster* subgroup. A** Summary of species recently invaded by *Shellder* and *Spoink*. The evidence of recent invasion is either based on time-series data (red) or a patchy geographic distribution of the TE (orange). **B** Schematic overview of the distribution of the involved species or species groups. **C** Reconstructed order of *Spoink*, *Shellder* and *P-element* invasions in the *melanogaster* subgroup. Arrows indicate likely horizontal transfer events and numbers provide the approximate year of the invasion.

*Shellder* and *Spoink* is very similar among the different species of the *melanogaster* subgroup, therefore we cannot infer the order of events from the phylogenetic tree alone (Fig. 4). To reconstruct the order of events, we had to draw cues from additional sources of information. First, the time-series data we compiled for several species offer valuable insights into the timing of the invasions and, consequently, the most likely sequence of horizontal transfer events. Our data suggest that *Spoink* invaded *D. melanogaster* between 1983 and 1993, prior to its invasion of *D. simulans* between 1995 and 2005. Therefore, a horizontal transfer of *Spoink* from *D. simulans* to *D. melanogaster* seems unlikely, although it cannot be completely ruled out due to the possibility of a considerable lag time between the initial transmission of a TE and its subsequent spread in populations. Second, insights can be gained from the geographic distribution of the involved species (Fig. 5B). It is unlikely that *Shellder* directly infected *D. mauritiana* from a species of the *cardini* or *repleta* group, since the ranges of these species do not overlap. Species of the *cardini* and *repleta* group are endemic to South America, while *D. mauritiana* is found on the island of Mauritius[101]. Therefore, an additional species bridging the geographic gap between Mauritius and South America must have acted as an intermediate host. Third, additional evidence can be obtained from the number of TE variants shared between species. Our approach identifies SNPs in TE sequences, where a SNP refers to a variant among the dispersed copies of the TE. If such segregating SNPs are found in two species, it indicates that both species share the same TE variants. The presence of multiple TE variants in several species suggests either multiple horizontal transfer events or a single event involving the transmission of several TE variants. However, we were only able to investigate segregating SNPs for *Shellder*, since the presence of degraded *Spoink* fragments in many species (likely from an ancient invasion of a TE with some sequence identity to *Spoink*) confounds this analysis (Fig. 4A; ref. 51). Interestingly, we found several segregating *Shellder* SNPs shared between *D. simulans*-*D. mauritiana* and *D. simulans*-*D. sechellia*, but only one between *D. mauritiana*-*D. sechellia* (supplementary Table S13). This suggests that *Shellder* was transmitted between *D. simulans* and *D. mauritiana* (*D. sechellia*) but not between *D. mauritiana* and *D. sechellia*. This is consistent with the geographic distribution of *D. mauritiana* and *D. sechellia* being endemic on different islands in the Indian Ocean. Furthermore, given that several segregating SNPs are shared between *D. simulans* and *D. mauritiana* (*D. sechellia*) and that these species are hybridising[92,94,95] it is likely that hybridisation led to the the transmission of several *Shellder* variants among these species. No shared SNPs were found between *D. teissieri* and any of the other analysed species, suggesting that a single horizontal transfer event triggered the *Shellder* invasion in *D. teissieri* (Supplementary Table S13). By integrating all possible sources of

information (species distributions, timeline of the invasions, phylogenetic trees, shared and segregating SNPs), we propose the following order of events (Fig. 5C).

*Spoink* was most likely transferred from the *willistoni* group to *D. melanogaster*, where it spread in worldwide populations between 1983 and 1993 (Fig. 5C). Next, *D. simulans* populations were invaded between 1995 and 2005 following a horizontal transfer from either *D. melanogaster* or from a species of the *willistoni* group. An alternative scenario involves *D. simulans* transferring *Spoink* to *D. melanogaster*, but we consider this less plausible as our time-series data suggest that *D. simulans* was invaded after *D. melanogaster* and because *D. melanogaster* does not carry *Shellder* (which is present in *D. simulans*). Lastly, *D. simulans* infected *D. mauritiana* and *D. sechellia* probably through hybridisation, resulting in *Spoink* spreading in *D. mauritiana* and *D. sechellia* after 1995. However, the presence of *Spoink* in *D. sechellia* strain *sech25* is in conflict with this scenario. Therefore, we cannot rule out that *Spoink* was already present around 1981 in some *D. sechellia* strains. Alternative explanations could be an incorrectly inferred sampling date of this strain (inferred from refs. [102],[103]) or issues during stock keeping. Another possible scenario involves *D. melanogaster* transferring *Spoink* to *D. sechellia* and *D. mauritiana*. However, since *D. melanogaster* does not hybridise with either of these species[95], this scenario is considered less credible than a transfer from *D. simulans* via hybridisation.

*Shellder* most likely followed a similar path as *Spoink*, with the exception that *D. melanogaster* does not carry the TE (Fig. 5C). A horizontal transfer from the *cardini* or the *repleta* group triggered the *Shellder* invasion in *D. simulans*, which would then have passed it to *D. mauritiana* and *D. sechellia* by hybridisation (these species share several segregating *Shellder* SNPs; see above). *D. teissieri* was likely infected by *D. simulans* given that the range of *D. sechellia* and *D. mauritiana* does not overlap with *D. teissieri*. A single horizontal transfer event presumably triggered the *Shellder* invasion in *D. teissieri* (no segregating SNPs are shared between *D. teissieri* and any of the other infected species; Supplementary Table S13).

As a reference, we also analysed the *P-element*, for which the invasion history in the *melanogaster* subgroup has been detailed in previous studies[42,48,50,60]. Based on our data, we infer that the *P-element* spread from the *willistoni* group to *D. melanogaster*, invaded worldwide populations of *D. melanogaster* around 1970, and then infected *D. simulans* populations approximately 50 years later (Fig. 5C). Other species in this group, such as *D. mauritiana* and *D. sechellia*, have not yet been invaded by the *P-element*. This scenario is in good agreement with previous studies[42,48,53,60].

In summary, drawing on many different sources of information, we were able to reconstruct the order of events that led to the invasion of *Shellder* and *Spoink* in multiple species of the *melanogaster* subgroup during the last 50 years. This allows us to highlight similarities and differences between the invasions of the different TEs. One notable common feature is that all three recent invasions in multiple species of the *melanogaster* subgroup, i.e., *Spoink*, *Shellder* and the *P-element*, were triggered by a horizontal transfer from a species distributed in South America. All three invasions described here only became possible after *D. melanogaster* extended its range into the Americas ~200 years ago[59]. Of note, it is also likely that *Shellder* infected the two *Anastrepha* species in South America (all involved species solely occur in South America). Finally, we suggest that cosmopolitan species may act as global hubs, distributing invading TEs between species separated by wide geographic gaps, such as South American species and species endemic to islands in the Indian Ocean.

## Discussion

In this work, we discovered that the retrotransposons *Shellder* and *Spoink* invaded several species of the *melanogaster* subgroup within the last 50 years. Typically, the horizontal transfer of a TE is inferred

from three distinct lines of evidence: (i) a patchy distribution of the TE among closely related species, (ii) a phylogenetic discrepancy between the TE and the host species and (iii) a high sequence identity between the TE of the donor and recipient species, often quantified by the synonymous divergence of the TE[44,114,115]. All three support the recent invasion of *Spoink* in *D. melanogaster*[51]. In addition, we provide further substantiation for the recent *Spoink* invasion: we show that *Spoink* insertions were most likely absent in worldwide *D. melanogaster* populations before 1983 but present in all populations after 1993 (see also Supplementary Fig. S7). Similarly, we present compelling support for the invasions of *Shellder* and *Spoink* in multiple species of the melanogaster group. We observed a patchy distribution of both TEs among drosophilids (Fig. 4A), a phylogenetic discordance between the TEs and their host species (Supplementary Fig. S14), and a lower divergence of the TEs compared to host genes (Supplementary Fig. S15). Furthermore, we provide direct evidence for the recent invasions of these TEs. We show that *Shellder* and *Spoink* were largely absent from the investigated *D. simulans* strains sampled before 1995 but present after 2012. In addition, available data suggest that these two TEs were absent in *D. mauritiana* ~1980 but present ~2006. Our time-series data are reliable for inferring the spread of a TE, as demonstrated by the *P-element*. Based on our approach, we infer that the *P-element* spread in *D. melanogaster* ~1970 and in *D. simulans* ~2010, in good agreement with previous studies[42,48,50,60]. However, for *D. sechellia* and *D. teissieri* time-series data are not available. Nevertheless, we proposed that *D. sechellia* was recently invaded by *Shellder* and *Spoink*, and *D. teissieri* by *Shellder*, based on a geographically patchy distribution of these TEs, where some populations have a TE while others do not. Invasions will commence at the site of the horizontal transfer (likely South America for *Spoink*, *Shellder* and the *P-element*), and then gradually, due to migration, spread to more distant populations. A geographically patchy distribution is expected during the invasion. In agreement with this, Anxolabéhère et al.[50] observed that the *P-element* started to spread in populations from North America in the 1960's and that by the 1980's most populations had *P-element* insertions. Another case is the recent invasion of *copia* in *D. willistoni* which led to a patchy geographic distribution[118]. We also observed a transient geographically patchy distribution for *Spoink*, *Shellder* and the *P-element* between 1994 and 2016 (Fig. 1B). Thus, we argue that a geographically patchy distribution of a TE should be considered evidence of a recent invasion, likely triggered by a recent horizontal transfer.

An alternative hypothesis is that both TEs were vertically transmitted and then recently reactivated. This would imply that both TEs were consistently present in a few individuals of the invaded species and only recently spread to the rest of the population. However, this hypothesis has several issues: (i) we did not find the TEs in any sample prior to 1970 in *D. simulans*, prior to 1965 in *D. melanogaster*, and prior to 2000 in *D. mauritiana*, (ii) it cannot explain why the TEs were confined to a few individuals for many thousands years, but then spread to most investigated individuals at about the same time in several species during the last 50 years, (iii) it does not account for the discordance in the phylogeny of the host species and that of the TEs, (iv) it fails to explain the high sequence identity of the TEs between very distantly related species, as the TEs exhibit lower divergence than any of the host genes (Supplementary Fig. S15).

It is interesting that the geographically patchy distribution of *Shellder* (*Spoink*) in *D. sechellia* and *D. teissieri* is linked to population structure (Figs. 2D, 3C). This could just reflect that gene flow and the spread of a TE face similar physical obstacles. For example, the distribution of *D. sechellia* on different islands may restrict the migration of flies between the island populations, leading to the observed population structure at the genomic level. At the same time, the limited migration of flies between island may be an obstacle to the spread of the TE.

                                                                    

Although we have solid evidence supporting the invasions of *Shellder* and *Spoink* in multiple species of the *melanogaster* subgroup during the last 50 years, we caution that our estimates of the invasion time should be treated as approximations that may be off by several years. Furthermore, we note that the time of the horizontal transfer may have occurred earlier since there could be a lag-time between the transmission and the spread in natural populations. We also note that the horizontal transfer of both TEs could not be much older than 200 years, since *D. simulans* and *D. melanogaster* were absent in South America (i.e., the range of the putative donor species of *Spoink* and *Shellder*) before that time[57–59].

Strikingly, *Shellder* and *Spoink* are co-occurring in *D. simulans*, *D. mauritiana* and *D. sechellia*, where both TEs are either jointly present or absent in the investigated strains (exception being a single long-read assembly of *D. sechellia*). What could be responsible for such a close association between *Shellder* and *Spoink*? Since both TEs have different insertion sites, we could rule out that they are transposing as a unit in *D. simulans* and *D. mauritiana* (Supplementary Tables S5, S7; a long-read assembly having both TEs is not available for *D. sechellia*). It could be that *Shellder* and *Spoink* depend on one another for transposition, e.g., some enzymes produced by one TE could be required by the other. *env* glycoproteins generated by one TE (e.g., *Shellder*) could be utilised by the other TEs (e.g., *Spoink*) to enhance its transmission[119]. However, we can rule out the hypothesis that the presence of either TE is required for the spread of the other, because we found just *Spoink* in *D. melanogaster* and only *Shellder* in *D. teissieri*, demonstrating that both TEs may spread independently. This does not rule out that the products of one TE may support the spread of the other. Another possibility is that both *Shellder* and *Spoink* were transmitted to *D. simulans* from the same donor species. Our results suggest that *Shellder* and *Spoink* are derived from different species (*Spoink* from willistoni and *Shellder* from the cardini or repleta group) making this hypothesis unlikely. However, we cannot rule out that a species that has not yet been sequenced acted as the common donor of both TEs. It is most probable that the two TEs are associated by pure coincidence. Both *Shellder* and *Spoink* may have invaded *D. simulans* around the same time from different donor species in South America. Gene flow in *D. simulans* then led to the joint spread of both TEs. Later, both TEs infected *D. mauritiana* and *D. sechellia* by hybridisation, where gene flow again led to the joint colonisation of populations by both TEs.

It is interesting to speculate which TE invasions may follow. Initially, it is likely that *Shellder* and *Spoink* will spread throughout the populations of *D. teissieri* and *D. sechellia*, thus resolving the patchy geographic distribution. Given that *D. simulans* has the *P-element* and is hybridising with *D. mauritiana* and *D. sechellia*, which do not have it, we expect that the *P-element* will soon invade *D. mauritiana* and *D. sechellia*. Similarly, as *D. teissieri* harbours *Shellder* and is hybridising with *D. yakuba* (*Shellder* naive) we expect that *Shellder* will soon spread to *D. yakuba*. As *D. yakuba* is hybridising with *D. santomea*, it is feasible that *Shellder* will soon invade also *D. santomea*. Given that at least six different TE families were horizontally transmitted between *D. simulans* (having *Shellder*) and *D. melanogaster* (not having *Shellder*) during the last 200 years, we also expect that *Shellder* will spread in *D. melanogaster* in the next decades[41,51]. It is also feasible that *Spoink* and the *P-element* may soon invade *D. teissieri*, *D. yakuba* and *D. santomea*. It will be interesting to see whether any of these TEs will spread to more distantly related species of the *melanogaster* group, such as *D. erecta* and *D. orena*.

Our work demonstrates the cascading nature of TE invasions, where the infection of one species can result in the spread of a TE throughout a group of related species. In particular, cosmopolitan species can act as a vector between local populations that would not individually come into contact. Once a TE escapes its local host species by jumping into a cosmopolitan species, the TE may spread to geographically distant species, such as island endemic species. The three major properties of invasion cascades are therefore: (i) a TE can infect many related species, (ii) these infections can occur within a short period of time (a few decades), and (iii) species with distributions geographically very distant from the TE's origin can get infected. The invasions of all three TEs - *Spoink*, *Shellder* and *P-element*- were triggered by a horizontal transfer from a species in South America. These invasions became feasible only after *D. melanogaster* and *D. simulans* expanded their ranges into the Americas, likely due to human activity, ~200 years ago[57–59]. These range expansions not only introduced previously isolated species into contact but also increased the population sizes of invasive species and exposed them to novel vectors for horizontal transfer. This heightened the potential for additional transfers between species and may have triggered invasion cascades. It is estimated that more than 7000 insect species are now found outside of their native range[120,121]. Given that insect range expansions often lag behind those of plants, ~3,000 expansions of insects are expected to follow[121]. These shifts provide vast opportunities for novel horizontal transfers of TEs across insect species. Human activity contributes to these expansions both directly by introducing alien species through global trade and indirectly, by altering habitats via climate change[1–4]. Our findings suggest that the impact of range expansion on genome invaders extends beyond the species directly involved. The cascading effect of TE invasions may ripple across ecosystems, affecting even species endemic to isolated islands. As climate change and human movement continue to alter species ranges, it will be crucial to understand whether such invasion cascades could affect the evolution of global insect species.

## Methods

### Characterising *Shellder* and *Spoink*

We generated a phylogenetic tree based on the reverse transcriptase domain of *Shellder* and *Spoink*. Briefly, we picked several sequences from each of the known LTR superfamily/groups[74,122], performed a blastx search to identify the RT domain[123], a multiple sequence alignment with MUSCLE (v3.8.1551)[124], generated the trees with BEAST (v2.7.5)[125] and picked the maximum credibility tree with TreeAnnotator (v2.7.5)[125].

### Insertion location of *Shellder* and *Spoink*

Genes were annotated in individual *D. simulans* genomes using the reference genome of *D. simulans* (v. 2.02; Flybase) and liftoff 1.6.3[126,127]. The location of *Shellder* and *Spoink* insertions within or near genes was determined with bedtools intersect[128]. If a TE was inserted within 1 kb of the transcription start site of a gene, it was classified as inserted into a promoter. TE insertions were also visually inspected in IGV (2.3.35) to validate the results of bedtools intersect[129].

### Copy number estimates using short-read data

We investigated the abundance of *Spoink*, *Shellder* and *P-element* in multiple publicly available short-read data sets. We analysed 179 strains of *D. melanogaster*[52,90,130–133], 88 of *D. simulans*[60,69,75–79], 43 of *D. sechellia*[92,93], 12 of *D. mauritiana*[91,96–98], 13 of *D. teissieri*[93], 36 of *D. yakuba*[93,110], 4 of *D. erecta* and 1 of *D. orena*[111]. For an overview of all analysed short-read data, see Supplementary Tables S2, S3, S6, S8, S9, S10, S11. We trimmed the short reads to a length of 100 nucleotides, discarding any shorter reads. Subsequently, we merged the paired-end files, as the following analyses did not require paired-end information. The trimmed reads were mapped to a database consisting of the consensus sequences of *Shellder*, *Spoink* and *P-element*, and three single copy genes (*Antennapedia*, *Osi6* and *Wingless*; the corresponding orthologs have been used for each species) with bwa bwasw (version 0.7.17-r1188)[134]. We used DeviaTE (v0.3.8)[80] to estimate the abundance of the TEs. DeviaTE estimates the copy number of a TE (e.g., *Shellder*) by normalising the coverage of the TE by the coverage of the single copy genes. We also used DeviaTE to visualise the abundance

and diversity of the TEs as well as to compute the frequency of SNPs in TEs. We used ggplot2[135] to generate the plots.

## Copy number estimates in long-read assemblies

We investigated the abundance of *Spoink*, *Shellder* and the *P-element* in long-read assemblies. The 49 analysed *D. melanogaster* strains were described previously[51] (data from refs. [87–90]). The *D. simulans* strains *SZ45*, *SZ129*, *SZ232*, *SZ244*, *NP-15-062*, *MD251*, *MD242*, *NS137*, and *NS40* are described in ref. [43]. *MD106* was collected by Ballard in 2002 and has been deposited under PRJNA1077914. *D. simulans 14021-0251.006* was collected in Nueva, CA, in 1961 and its sequencing is described here[106]. *D. simulans w*[501] was collected between 1930 and 1960, and its sequencing and assembly is described here[79](PRJNA377886). The *D. mauritiana* strain 14021-0241.01 is described here[106] and *w*[12] here[71]. The R strains of *D. mauritiana* were collected in 2006 and kindly donated by M. Ramos-Womack, provided to B. Kim by C. Meiklejohn[136–138]. We sequenced and assembled these strains as described previously[43] PRJNA1077916. We identified insertions of *Spoink*, *Shellder* and *P-element* in these assemblies using RepeatMasker with a custom library consisting of the three TEs consensus sequences (open-4.0.7; -no-is -s -nolow)[139]. We solely counted insertions with >750 bp of length and <10% sequence divergence.

## small RNA analysis

We used previously published short-RNAs from *D. simulans* ovaries (ERR1821669)[81]. The adaptor sequence GAATTCTCGGGTGCCAAGG was trimmed with cutadapt 4.4[140]. The trimmed reads where mapped using novoalign[141] to the consensus sequences of TEs, including the sequences of *Shellder* and *Spoink*[122]. We used a Python script to compute the ping-pong signature and the distribution of the piRNAs along the TEs[22].

## TE insertions into *flamenco*

*flamenco* was annotated in each genome as described in ref. [43]. TEs were annotated in each genome with RepeatMasker[139] (open-4.0.7; -no-is -s -nolow). *flamenco* regions were inspected for *Shellder* and *Spoink* insertions and aligned with Manna (which aligns TE annotations)[84]. Lack of synteny between the TEs surrounding a *Shellder* insertion was considered evidence for independent *flamenco* insertion. The alternative would be that genome rearrangements occurred precisely on both borders of *Shellder* such that a single insertion event no longer contained any of its original neighbouring sequences.

## Population structure in *D. sechellia* and *D. teissieri*

To investigate the population structure of *D. teissieri* and *D. sechellia* strains with a PCA, we mapped the raw sequencing reads to the reference genome (*D. sechellia*: GCA_004382195.2, *D. teissieri*: GCA_016746235.2) with bwa-mem (v0.7.17-r1188)[134], called variants using bcftools (v1.17)[142] and performed PCA with plink2 (v1.90b5)[143]. To obtain a phylogenetic tree based on these variants, we converted the vcf file to the phylip format (https://github.com/edgardomortiz/vcf2phylip)[144] and generated a tree using BEAST (v2.7.5)[125].

## Shared TE SNPs

The allele frequencies for each position in *Shellder* were obtained from the DeviaTE output. SNPs were called at sites with a minor allele frequency of >0.1. To detect shared SNPs we intersected the coordinates of SNPs among pairs of species using R (v4.2.0)[145].

## Origin of horizontal transfer

We downloaded long-read assemblies of 266 drosophilids[106,112] and of 1225 insects reference genomes from NCBI. The insects genomes were found by filtering for "insects", "chromosome level" and "reference" at the NCBI database. The list of all analysed drosophilid and insect species, including the source, can be found in Supplementary Datasets S5, S6. We used RepeatMasker[139] (open-4.0.7; -no-is -s -nolow) to identify sequences with similarities to *Shellder* and *Spoink* in these genomes. Using a Python script, we identified the best hit for *Shellder* and *Spoink* in each assembly (i.e., the highest alignment score) and then estimated the similarity between this best hit and the TE using the equation $s = rms_{best}/rms_{max}$, where $rms_{best}$ is the highest RepeatMasker score (rms) in a given assembly and $rms_{max}$ the highest score in any of the analysed assemblies. A $s = 0$ indicates no similarity to the consensus sequence of the TE, whereas $s = 1$ represent the highest possible similarity. To produce the phylogenetic tree of the 266 drosophilids we used BUSCO[146] to extract the sequences of the identified 146 orthologous genes. With MUSCLE (v3.8.1551)[124] and RaxML (v8)[147] we generated gene trees for the orthologous genes. The gene trees were summarised with Astral (v5.7.8)[148]. To generate phylogenetic trees for the TEs, we identified *Shellder* and *Spoink* insertions in the assemblies of the 266 drosophilid species using RepeatMasker. We extracted the sequences of complete insertions (>80% of the length; two LTRs) from species having at least one full-length insertion using bedtools[128](v2.30.0). For each TE, a multiple sequence alignment of the insertions was generated with MUSCLE (v3.8.1551)[124] and a tree was generated with BEAST (v2.7.5)[125].

## Reporting summary

Further information on research design is available in the Nature Portfolio Reporting Summary linked to this article.

## Data availability

The newly assembled *D. mauritiana* strains are available at NCBI under accession number PRJNA1077916. Accessions for all genomic data analysed in this study are listed in S5, S6, and in the Supplementary Material. All analysis performed have been documented with RMarkdown and have been made available, together with the resulting figures in GitHub (https://github.com/rpianezza/DoubleTrouble, ref. [149]). Source data are provided in this paper.

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

## Acknowledgements

This work was supported by the National Science Foundation Established Programme to Stimulate Competitive Research grants NSF-EPSCoR-1826834 and NSF-EPSCoR-2032756 to SS and by the Austrian Science Fund (FWF) grants P35093 and P34965 to R.K. This work was also supported by the National Institutes of Health (R01-GM083300 and R01-HD108914) to E.C.L. and NIH core grant P30-CA008748 to Memorial Sloan Kettering Cancer Centre. All of the authors would like to thank Colin D. Meiklejohn for providing the *D. mauritiana* strains for sequencing and the Petrov lab at Stanford University for hosting the work of H. Gellert and B.Y. Kim. S.S. would like to thank T. Greives for *D. melanogaster* strains and P. Senn for inspiration. We thank M. Beaumont for the invaluable feedback on the paper. R.K., A.S., R.P. and A.H. thank all members of the Institute of Population Genetics for feedback and support.

## Author contributions

S.S. discovered the *Shellder* and *Spoink* invasions. R.K. and S.S. conceived this work. R.P., A.S., S.S., A.H. and R.K. analysed the data. H.R.G., B.Y.K. and E.C.L. contributed genome assemblies. R.K. and S.S. wrote the manuscript. A.S. and R.P. contributed to the writing.

## Competing interests

The authors declare no competing interests.
