## [Transparent Peer Review file · Nature Communications]

Double trouble: two retrotransposons triggered a cascade of invasions in *Drosophila* species within the last 50 years

Corresponding Author: Dr Robert Kofler

Version 1:

Reviewer comments:

Reviewer #1

(Remarks to the Author)

The present manuscript reports data from genome analyses showing the recent invasion of 3 or 4 species respectively of the *Drosophila melanogaster* subgroup by the Shellder and Spoink transposable elements (TEs). The results also show that genomic elements related to Shellder and Spoink are present in other drosophilid species, with varying degrees of similarity. The authors suggest a horizontal transfer of Shellder and Spoink, less than 50 years ago, from the willistoni, cardini and repleta groups to species of the melanogaster subgroup and conclude that there has been a cascade of TE invasions over the last 50 years, probably favoured by human activity. They suggest species hybridisation as a possible mode of horizontal transfer.

The manuscript is based on bioinformatics analyses of 266 long-read genome assemblies (209 *Drosophila* genomes, 49 genomes of other drosophilid species), of short-read genomic data sets from 380 strains from the melanogaster subgroup (183 of *D. melanogaster*, 88 of *D. simulans*, 43 of *D. sechellia*, 12 of *D. mauritiana*, 13 of *D. teissieri*, 36 of *D. yakuba*, 4 of *D. erecta* and 1 of *D. orena*), and of 1225 insect reference genomes from NCBI. It also presents results from the analysis of a sample of small-RNA sequencing data from ovaries of a *D. simulans* strain.

The invasion of *D. melanogaster* by Spoink has already been published in the cited publication by Pianezza et al., 2023. In the latter publication, the presence of Spoink in *D. sechellia* and *D. simulans* has also been shown. The presence in *D. willistoni* is implied since Spoink's internal sequences are 99.7 % identical to the *D. willistoni* TE Gypsy-78_DWil. What's new here for Spoink is its presence in *D. mauritiana* and the occurrence of related copies in drosophilid species outside of the melanogaster subgroup. The presence of Shellder in *D. simulans* and *D. mauritiana* has been published in the cited paper by Ding et al., 2016. A TE showing high similarity with Shellder, Gypsy-29_DWil, had been identified by others in *D. willistoni*. What is new in this manuscript is the extensive search for Shellder and Spoink in other drosophilid species. While invasion of *Drosophila* species by 7 other TEs over the last 200 years has been reported (references in Pianezza et al., 2023), and 101 putative events of horizontal transfer have been proposed in *Drosophila* for 21 different elements (Loreto et al., 2008), the novelty here is that the authors suggest horizontal transfer and invasion by Shellder and Spoink during the last 50 years.

Overall, this manuscript contains very interesting results and builds a nice story of recent species invasion by two transposable elements in *Drosophila*, it is of significance to the field. Meanwhile, the conclusions about horizontal transfer need to be reconsidered in depth, and the title needs to be changed, replacing "horizontal transfers" with "invasions" (see my comments below).

Major comments:

Inferring horizontal transfer is quite tricky. The methods for inferring horizontal transfer are well described in the cited publication by Loreto et al. 2008. It is even more difficult to estimate when a putative horizontal transfer took place and from which donor species. The results presented in this manuscript are very interesting and there is a large amount of data, but the conclusions about horizontal transfer drawn from these results are not convincing. The statement in the title that "two retrotransposons triggered a cascade of horizontal transfers in *Drosophila* species within the last 50 years" is clearly overstated.

Horizontal transfer between species over the last 50 years should create copies of the respective transposable elements that are almost identical to the donor element. This is only the case within the group of "invaded species" (Figure 4A), i.e. for Shellder in *D. simulans*, *D. teissieri*, and *D. mauritiana*, and for Spoink in *D. melanogaster*, *D. simulans*, *D. sechellia*, and *D. mauritiana*. In the other species, the "candidate donor species", the similarity is much lower. A good example of recent horizontal transfer, between 1950 and 1980, is the P element, which is 3 kb long and differs in just one nucleotide between *D. melanogaster* and *D. willistoni* (Daniels et al., 1990). Also, the Bayesian trees of Shellder and Spoink insertions in Figures 4B and 4C do not support very recent horizontal transfer in the last 50 years between one of the species of the melanogaster subgroup and a species not belonging to this group. This means that the horizontal transfer from the "candidate donor species" to the "invaded species" more likely is a quite ancient event and that the scenario presented in Figure 5C, and especially the dating of horizontal transfer events, is extremely unlikely. It can clearly not be stated that all the horizontal transfer events suggested here have occurred in the last 50 years. It is more appropriate to conclude that the donor species or population from which Shellder and Spoink invaded the melanogaster subgroup is not present in the 266 samples analysed here, as none of the "candidate donor species" in Figure 4 have copies of Shellder or Spoink that are nearly identical to copies of the "invaded species". A possibly functional TE related to Spoink, Gypsy-78_DWil, had been identified by others in *D. willistoni*. Gypsy-78_DWil shows 99.73% similarity with Spoink for internal sequences, but only 93.54% identity for LTRs (Pianezza et al., 2023). No possibly functional Shellder copy, having full coding capacity and identical LTRs, is reported for the putative donor species in this manuscript. Consequently, the donor species cannot be clearly identified and the time of horizontal transfer cannot be determined. Only eventual horizontal transfer events between the "invaded species" may be relatively recent, between species of the melanogaster subgroup, *D. melanogaster*, *D. simulans*, *D. sechellia*, *D. mauritiana* and *D. teissieri*. So the hypothesis of a recent "cascade of horizontal transfers" is not supported by the data. In addition, the events may not be independent, as shown in Figure 2, which suggests that Spoink and Shellder were transferred together. And finally, the divergence of the Shellder and Spoink copies in the different genomes (Figure 4B, C) rather suggests that there is a timelaps between the moment of horizontal transfer and the time of invasion. It's important not to confuse the two. It is more likely that the horizontal transfer between species occurred some time ago already, more than 50 years ago, and that the invaded population only recently spread and 'contaminated' other populations of the same species around the world. The time of invasion is clearly not the time of horizontal transfer. This is mentioned in the manuscript: "... there could be a considerable lag time between the initial transmission of a TE and its spread in populations." (line 362) but not really considered when concluding for the time of horizontal transfer events. This should be considered all over the manuscript and discussed. The title needs to be changed, as only invasion has occurred recently with any certainty, and not horizontal transfer.

lines 23-25:

"Our work reveals that cascades of TE invasions, likely initiated by human-mediated habitat expansions, could have a profound impact on the genomic and phenotypic evolution of many geographically dispersed species." This generalisation seems exaggerated, as these observations have only been made for drosophilid species and the mode of horizontal transfer is not known.

line 122:

It would be important to note here that Gypsy-29_DWil is a TE that has been identified in *D. willistoni* and discuss this important point.

line 124:

"We found that gypsy-29_DWil is identical (99.9%) to Shellder, a TE previously described by Ding et al. [2016]." The authors indicate 99.9% identity of Shellder with Gypsy-29_DWil. This is not correct. Comparing Shellder, published by Ding et al., 2016, Genbank record KX196449.1, and Gypsy-29_DWil (from Repbase), there is only 97% identity between the internal sequences and 88% identity between the respective LTRs. If Gypsy-29_DWil were 99.9% identical to Shellder from *D. simulans*, this would have been the only strong argument in favour of a recent horizontal transfer from *D. willistoni* to the melanogaster subgroup. But this is clearly not the case.

line 141-143:

Although 88 lines have been analysed for *D. simulans*, many of them were collected at the same time in the same country, possibly even in the same locality: 19 of them were collected in Kenya in 2018, 11 in Kenya in 2006, 12 in Madagascar in 2002 etc., which means the sampling is in fact quite limited from a statistical point of view. Thus, the statement "This suggests that our method accurately captures the timing of TE invasions" (line 149) is clearly exaggerated.

lines 151-153:

"The presence of Shellder and Spoink in one strain collected 1975 in Kenya (Ken75, SRR22548178) may be due to contamination of the strain."

This seems purely speculative and not based on any data, e.g. contaminated and non-contaminated samples from this Kenya 1975 strain. Then, how can the authors say that this strain is contaminated and not the others? It is entirely possible that the 1975 Kenyan sample came from a population in a different region of Kenya to the other samples. Did the authors check this? It is much more likely that the invasion had already begun in 1975. Importantly, the 1975 Kenyan sample is not shown in supplementary fig. S3 while the authors refer to this Figure (line 155). The corresponding results need to be shown and a non-contaminated sample of this 1975 Kenyan strain would be a good thing to find.

Although many genomes of different species have been analysed, the data are not always sufficient to justify the conclusions drawn. It would be enough to find a single *Drosophila* somewhere, among museum specimens for example, from before 1974 and having Shellder and/or Spoink to invalidate the hypothesis of horizontal transfer and invasion only after 1974. It is impossible to determine when horizontal transfer took place, as there are not enough genomic sequencing

data for ancient times. It is therefore clearly an overstatement to claim that horizontal transfer and invasion have occurred in the last 50 years, and not before. Nevertheless, the results presented strongly suggest that the invasion of the studied populations is recent. An interesting question is then to determine which population was the first to be contaminated and where it came from. The authors attempt to answer this question, but here too we come up against a lack of available samples: the 88 genomes analysed for *D. simulans* in Supplementary Table 2, for example, come from only 22 different regions; only 5 genomes come from *D. simulans* captured before 1974, and of these, 2 correspond to the same strain w501 (Figure 1, supplementary Table 2). The same is also true for the other species, where many samples originate from the same regions (see suppl. Tables 7-10). Thus, we cannot be sure that Shellder and Spoink were absent from a given species before 1974.

Shpak et al., 2023 sequenced 25 genomes from museum specimens of *Drosophila melanogaster* ranging from the early 1800s to 1933 ("Genomes from historical *Drosophila melanogaster* specimens illuminate adaptive and demographic changes across more than 200 years of evolution", PLoS Biol 21: e3002333). As I could not find a file with the analysed short-read genomic data sets, I could not see all the strains that were used in this study. Such a list should be joined to the supplementary. Did the authors study the genomic data from Shpak et al. 2023? This publication is not listed in the references.

lines 159-161:

" ... worldwide populations prior to 1994 ..."

The term " worldwide populations" suggests that many different populations captured at many different places "worldwide", all over the world, were used in this study. This is not the case. There are only 13 genomes from *D. simulans* captured prior to 1994 that have been analysed and these originate from 7 to 9 different regions (supplementary table 2).

Figure 4:

There is some incongruence between Figures 4A, 4B, 4C and the list of analysed data sets in supplementary file S1. For example, there are only 7 data sets for *D. melanogaster* in supplementary file S1 while there are 9 in Figure 4A and 17 data sets for *D. melanogaster* in Figure 4C. To be able to fully comprehend these data, the source data in a supplementary table would be required. It would also be necessary to give the results and references for all analysed data sets.

Supplementary file S1:

35 of the 266 data sets in supplementary file S1 are not yet publicly available (annotated "PENDING" and "xxx ... Sarah"), and 29 of these data sets come from other authors. Will these source data be made accessible and when? It would also be necessary to indicate the full species name for each data set.

line 323:

" ... we found insertions with a high similarity to Spoink and the P element in species of the willistoni group (fig. 4A; supplementary fig. S12)."

The term "high similarity" is exaggerated and not precise, unless there are other data that are not shown. The similarity seems to be less than 97%, which is not "high" for a hypothetical horizontal transfer less than 50 years ago. The similarity between Spoink and Gypsy-78_DWil from *D. willistoni*, high for internal sequences, relatively low for LTRs (see above, Pianezza et al., 2023), might be discussed here.

line 327:

"... insertions with a high similarity to Shellder were found in species of the cardini and the repleta groups"

As before, the term "high similarity" is exaggerated and not precise. It seems to be less than 97 or 98%, which is not "high" for a hypothetical horizontal transfer less than 50 years ago.

line 338:

"Spoink insertions of the melanogaster subgroup are nested in insertions from the willistoni group (fig. 4B)"

What is meant by "nested" here? Spoink copies of the willistoni group appear to be on distinct branches to copies of the melanogaster subgroup. Please clarify this important point.

"fig. 4B" should be replaced with "fig. 4C" for Spoink.

In addition to figure 4, it would be useful to include a table with the species in each group (repleta, cardini, saltans, willistoni, melanogaster) and the source data from figure 4A in a table.

line 341:

"Shellder insertions of the melanogaster subgroup are nested within species of the cardini and repleta groups, suggesting ..."

As before, what is meant by "nested"? Shellder copies of the repleta and cardini groups appear to be on distinct branches to copies of the melanogaster subgroup. Please clarify this important point.

It would be useful to give the shortest phylogenetic distances between copies of putative donor groups and the invaded groups and compare these to the longest phylogenetic distances within each group. The data in Figures 4B and 4C alone, do not support recent horizontal transfer, less than 50 years ago, between the "candidate donor species" and the "invaded species".

Line 435:

Loreto et al., 2008, should also be cited for the three methods.

lines 438-444:

These conclusions about presence/absence of Shellder and Spoink in the respective species should be tuned down because of the relatively low number of ancient samples.

lines 471-484:

It is not clear why the authors do not consider the possibility of a joint transfer of both Spoink and Shellder from the same donor for transfer within the melanogaster subgroup, eventually coupled to high copy number for both TEs and transposition of both TEs in the invaded populations, increasing their copy numbers. Please clarify.

Minor comments:

Abstract, line 18:

The authors propose hybridisation between species as the mode of horizontal transfer in the abstract. But hybridisation is only one of the multiple hypotheses that could explain horizontal transfers (see the cited Loreto et al., 2008). There is no strong argument in favour of species hybridisation. "Likely" should be replaced with "Possibly".

line 35:

TEs cannot be compared to extracellular disease-causing pathogens simply by describing them as "intragenomic parasites". TEs are not viruses. Please tune down.

line 61: Also cite Desset et al., 2003, for flamenco here, as "COM" and "flamenco" correspond to the same piRNA cluster.

line 78:

"Given the small number of known TE families in *D. melanogaster* (124) and the high number of TE invasions in the last 200 years, it is unlikely that this rate of invasions is the normal 'background' level of introduction.

The rate of invasion cannot be estimated solely on the basis of the number of known TE families.

There are at least 180 known TE families in *D. melanogaster* in the databases (Dfam, Repbase). Please check. In addition, some TEs have been published but do not appear in TE databases (like Spoink and Shellder). Furthermore, to date, only a few strains of *D. melanogaster* have been extensively studied for their TE content and the number of *D. melanogaster* genomes from around the world remains relatively small. Therefore there are probably more TEs to be discovered. The *D. melanogaster* genome is also full of dead transposable elements that invaded the species in the past and we do not know how long they may remain active. It is therefore possible that exchanges between populations and subsequent invasions are frequent in *Drosophila*. More data, particularly from old museum specimens, are certainly needed to estimate the rate of invasions at different periods in the history of *Drosophila*.

line 122:

"... second TE annotated as gypsy-29_DWil (from the repeat library of [Chakraborty et al., 2021]) ..."

The citation is not quite correct. Chakraborty et al., 2021 did not identify or describe any *D. willistoni* TEs. The "Gypsy-29_DWil" TE has been identified in *D. willistoni* and submitted to Repbase records by Jurka J. and Kohany O. in 2011. The publication cited here should refer to Repbase: Repbase, <https://www.girinst.org/>, Jurka J, Kapitonov VV, Pavlicek A, Klonowski P, Kohany O, Walichiewicz J. Repbase Update, a database of eukaryotic repetitive elements. *Cytogenet Genome Res.* 2005;110(1-4):462-7. doi: 10.1159/000084979. PMID: 16093699.

line 133:

"Spoink belongs to the gypsy/mdg3 family (members do not have env ..."

TEs that do not have an env are not necessarily of the same clade and I have no knowledge of a "gypsy/mdg3 family".

Please check the names of the clades. The Micropia/mdg3 clade has been described by Malik and Eickbush, 1999, *J.Virol.* 73:5186-5190 (see also <https://gydb.org/index.php/Ty3/Gypsy>).

line 134:

"In contrast to Spoink, Shellder may thus generate virus-like particles that could infect other species or tissues (e.g. the germline)."

The authors suggest here that Shellder could be a retrovirus. Until now, infectious virus-like particles involved in horizontal transfer between species have been suggested but there is no clear evidence for this mode of horizontal transfer in *Drosophila*. This should be clarified.

lines 136-137:

"Shellder and Spoink are both biased towards intergenic insertions, though Spoink inserts into introns more frequently than Shellder (30% versus 20%). Neither TE frequently inserts into other genic features, in contrast to the P-element where 42% of insertions are in promoters (supplementary fig S2)."

This statement may be incorrect as the data in supplementary fig S2 were apparently not normalized to the global size (or genomic fraction) of the respective regions in the genome. For example, promoters represent a very small fraction of the genome. Therefore, the probability of insertion into promoters is rather low compared to other regions. Idem for the legend of fig S2 where it is said that "Shellder has a pronounced insertion bias into intergenic regions."

The data in supplementary fig S2 concern only *D. simulans*. This must be specified.

Figure S2 concerns results from 10 (for Shellder and Spoink) or 3 genomes (for the P-element). The standard deviation between genomes should be shown in the figure.

What is meant by "other genic features" ? Please specify.

line 183:

"If Shellder is a somatic piRNA cluster ..."

Shellder is not a piRNA cluster but a transposable element.

line 199:

"Furthermore, Shellder and Spoink are controlled by the piRNA pathway in *D. simulans*."

There is no clear evidence in the manuscript that Spoink is controlled by the piRNA pathway in *D. simulans*.

line 202:

"Shellder is absent in *D. melanogaster*"

Even if 183 genomes, corresponding to 183 laboratory strains or wild-caught strains, often derived from a single female, have been analysed here, it is still possible that other *D. melanogaster* strains or populations exist that have Shellder. In addition, there are only 2 very recent strains, collected in 2015 and 2018, among the 183 strains. "Shellder was not found in *D. melanogaster*" would be a better title for this section.

The same considerations also apply to the "Shellder, but not Spoink, invaded *D. teissieri*" section (line 263), for example, where 15 strains were analysed, only 8 of which were certainly recent (the Bioko strains).

Ditto, "Shellder and Spoink are absent in other species of the melanogaster subgroup" (line 292) should be reworded to "Shellder and Spoink have not been found in other species of the melanogaster subgroup".

Figure 4, legend:

"full-length insertions (> 80% of length)"

80% is not "full-length", only 100% can be considered as "full-length". LTRs are often found different between related TEs. Do the authors mean full-length, i.e. 100% of length, alignment for the internal sequences only? Please specify or reword to avoid the term "full-length". The same goes for other figure legends using the same terms.

The legend of Figure 4B refers to both 4B and 4C.

Replace "266 *Drosophila* species" with "266 drosophilid species" and verify whether all 266 drosophilid species or the 209 *Drosophila* species are shown in Figure 4A.

Supplementary Figure 8:

"... hits with less than 750bp length or more than 10% divergence were removed..."

Does this mean that a fragment of e.g. 1kb length and having 9% divergence from the respective TE was considered as a "copy" here? In this case, it would be suitable to show also the copy number of full-length TEs with less than 2% divergence from the reference element for example to have an idea of TEs that potentially transposed less than 50 years ago. Indeed, ancestral copies of *Drosophila* TEs, which are not functional anymore (mutated ORFs) often show between 5 and 10% divergence, or even less, with the functional TE. Such ancestral copies are mostly estimated being older than 50 years. The literature about the rate of mutations affecting TEs in *Drosophila* should be studied and considered here.

Supplementary Figure 13, legend:

"... in 1226 long-read assemblies of insect species"

Are the authors sure that all 1226 assemblies are long-read assemblies, i.e. from Nanopore or PacBio sequencing ?

The source data should be made available to identify the species shown in figure S13A.

line 297:

"... infected by Shellder ..."

Shellder is not a virus. The term "invaded by Shellder" should be preferred.

line 300:

"were also absent from other species" should be replaced with "were not found in other species", given the low number of samples analysed for these species.

line 307:

"We thus concluded that Shellder and Spoink did not yet invade *D. yakuba*, *D. santomea*, *D. erecta*, and *D. orena*."

This conclusion is overstated given the low number of samples analysed for these species and identical geographic origin for many of them (e.g. 29 samples for *D. yakuba* originate from Sao Tomé).

Supplementary Table S9:

Replace "36 analysed *D. yakuba* strains" with "38 analysed *D. yakuba* strains".

line 464:

"multiple species of the melanogaster subgroup" refers to 3 or 4 species, replace "multiple" with "several" for example.

(Remarks on code availability)

Reviewer #2

(Remarks to the Author)

Scarpa et al. presented a work that examined a recent spread of retrotransposons in *Drosophila* species.

They focused on two gypsy LTR retrotransposons, Shellder and Spoink and examined the copy number in several *Drosophila* species, in particular, species from the melanogaster group, for which strains that were isolated across many years from different locations have been sequenced.

Authors found that both Shellder and Spoink were found only in some, but in all assemblies of the same species, and from that concluded that this is evidence of a recent invasion.

major comments

I appreciate that authors examined in-depth the copy number, sequence divergence, and geographical and chronological spread of these elements in several species.

The sporadic presence of a particular TE can indicate either a recent introduction of the TE into a subpopulation of the species (source of which could be a horizontal transfer), or a recent activity of a TE insertion that already existed within the species.

The paper entirely focused on the possibility of the former. However the latter scenario could also create a similar distribution pattern. The paper should investigate or at least discuss this alternative scenario.

The sequence divergence of these elements, for example, of Shellder as shown in Figure 4B is largely congruent with the phylogeny of species, which implies that the element has been vertically inherited in a large picture.

Authors focused on the connection between elements from the repleta group or cardini group that are closely related to elements found in the melanogaster group species.

This is not easily captured from the text nor from the figures.

I initially misread that the horizontal transfer occurred from a willistoni group species to a melanogaster group. But this is hardly possible given that the similarities between the willistoni Shellder elements and all the melanogaster Shellder elements are generally low (~80% according to Figure 4B).

I thank authors for putting details of genome assemblies used for the analyses and individual insertions in the supplemental tables.

By manually downloading the genomic sequences and clustering the open reading frames (ORFs), I could confirm that *Drosophila Mauritiana* R31 (GCA_039654645.1_ASM3965464v1) carries multiple copies of Shellder that are highly similar at the amino acid sequence level to the Shellder elements in *Drosophila anceps* (GCA_035045945.1_ASM3504594v1). Given that the melanogaster group and the repleta group separated more than 30 million years ago, this is a strong evidence for a horizontal transfer event.

However, I did not see intact copies of Shellder in the *Drosophila similans* genomes, such as, the sz232 and MD242 strains, which authors claimed carry multiple insertions.

These include their top hits "tig00000792 pilon pilon 450324 456077 +" and "tig00000776 pilon pilon 2683380 2689137 C" in the sz232 strain (supplemental table 1).

This is misleading because fragmented copies decay rapidly and affect tracking of the evolution of these elements between species.

Please provide information of where the intact copies of each assembly can be found.

Along the same line, by simple blast search, I did not find any copies of Shellder with all three ORFs in dsim_28_F0 (GCA_029774875.1_ASM2977487v1) or dsim_29_F0 (GCA_029774795.1_ASM2977479v1).

Please provide more detailed information that formed the basis of supplementary Figure 4.

Furthermore, I also failed to find intact copies in the *Drosophila cardini* genome (GCA_018903735.1_ASM1890373v1), in which authors found elements of a high similarity to the melanogaster Shellder (supplementary figure 12).

Overall, please clarify, wherever relevant, the exact region of TEs that was used to measure similarities between elements, and whether the insertions carry at least uninterrupted ORFs.

Without those data available, I cannot evaluate the findings made in the paper.

minor comments

1. Please make available the multiple sequence alignments of the RT domains used to construct the phylogenetic trees.

2. I cannot fully interpret the similarity plots.

What does similarity of <0.1 in a genome mean? Does that mean the most similar element in the genome has only such a little similarity to the TE in question? It is unlikely given that gypsy elements are highly abundant in most *Drosophila* genomes.

3. RepeatMasker uses many TE sequences that are similar to each other. This means that the lack of hits from a particular TE does not always mean that the genome has no similar elements. The examined genome may have hits from a very similar but different TE.

Please consider directly running blastn/tblastn of the TEs in question to rule out this possibility.

(Remarks on code availability)

Reviewer #3

(Remarks to the Author)

In the manuscript entitled "Double trouble: two retrotransposons triggered a cascade of horizontal transfers in *Drosophila* species within the last 50 years", the authors demonstrated the cascading invasions of two retrotransposons, Shellder and Spoink, among the worldwide *Drosophila* subspecies. To this end, they mined hundreds of short-reads and long-reads sequencing data from the time-series and geographic diverse *Drosophila* specimens. Their results reveal that the spread of these two retrotransposons was likely facilitated by human-mediated habitat expansion. Overall, this research is supported by extensive data from long-read assemblies and historical strains, providing a robust timeline for the invasion events.

1. In Figure 1, it would enhance the understanding of the invasion events if the authors could provide the chromosome ideograms displaying the locations of either full-length or fragment TEs overtime.

2. The co-occurrence of Shellder and Spoink in certain species is interesting. It would be beneficial if the authors could provide a more definitive explanation.

(Remarks on code availability)

Version 2:

Reviewer comments:

Reviewer #1

(Remarks to the Author)

The authors addressed all my concerns and changes were made to the manuscript accordingly. I'm entirely satisfied.

Only 3 minor changes remain:

Lines 325 and 331:

Please define the abbreviation 'SW' in the manuscript or replace it with 'Smith-Waterman'.

Line 521:

replace "repletagroup" with "repleta group"

Line 524:

replace "than" with "then"

Reviewer #2

(Remarks to the Author)

I thank the authors for addressing my questions point by point, with additional analyses and figures included.

I also thank that the authors made the raw data available in the git.

The authors addressed most of my questions. My major remaining question is the time scale of the horizontal transfer events.

That shellder/spoink are only found in recently collected melanogaster group strains, but not in earlier collections, suggests a recent introduction of these elements.

On the other hand, many of the insertions that the authors analysed and found most homologous to the consensus sequences in their respective genome assemblies do not code fully intact open reading frames. This means that despite the fact that there are multiple copies, none are currently active in those genomes.

This includes Shellder in *Dsim sz232* and Shellder in *D.cardini* (GCA_018903735.1).

I thank authors for making the genomic sequences of the top hits available in the git. Although they contain several long ORFs, none of them appeared to have continuous POL ORFs.

I find it remarkable that the horizontal transfer events, which are presumably rare, introduced these elements into multiple different species, each of which led to expansion in their respective genomes, and then to demise and a partial degradation within a span of 50 years. We don't encounter the same extent of changes in transposon insertions in the *Drosophila* genome in laboratory settings.

Because of this extraordinary sequence of events, I suggest authors explaining the full account of these events in the manuscript including the demise of these elements.

Especially, the current manuscript lacks information of whether the elements are currently transposing (namely the presence of fully intact copies) in which genomes.

Thank you for showing the mapping of short reads from dsim_28_F0 and dsim_29_F0 onto the transposon sequences. I presume that the 88 Dsim genomes that were analysed in Supplementary Figure 4 are all short reads. Please clarify this in the figure legend because in some cases, long-reads were used for Dsim strains. The use of short reads brings a question of the sensitivity. The copy number estimation may be affected by the sequencing depth as well as the type of sequencing method (single read vs paired end, read lengths, etc). Please could you comment on this potential confounder?

Authors mentioned that they used only the three relevant TEs as baits for RepeatMasker searches. Please state this more clearly in the method section.

Reviewer #3

(Remarks to the Author)

The authors have addressed all my questions. I appreciate the detailed explanation regarding the co-occurrence of Shellder and Spoink in the discussion. I also agree with the authors that the current ideogram data may not be particularly helpful for the audience. I have no further questions.

Version 3:

Reviewer comments:

Reviewer #2

(Remarks to the Author)

Authors addressed all of my questions.

I appreciate the efforts and the openness by the authors throughout the review process.

We thank the Reviewers for the effort and the detailed comments. We aimed to address all comments carefully.

Robert Kofler and Sarah Signor on behalf of all authors

REVIEWER COMMENTS

Reviewer #1 (Remarks to the Author):

The present manuscript reports data from genome analyses showing the recent invasion of 3 or 4 species respectively of the *Drosophila melanogaster* subgroup by the Shellder and Spink transposable elements (TEs). The results also show that genomic elements related to Shellder and Spink are present in other drosophilid species, with varying degrees of similarity. The authors suggest a horizontal transfer of Shellder and Spink, less than 50 years ago, from the *willistoni*, *cardini* and *repleta* groups to species of the *melanogaster* subgroup and conclude that there has been a cascade of TE invasions over the last 50 years, probably favoured by human activity. They suggest species hybridisation as a possible mode of horizontal transfer.

The manuscript is based on bioinformatics analyses of 266 long-read genome assemblies (209 *Drosophila* genomes, 49 genomes of other drosophilid species), of short-read genomic data sets from 380 strains from the *melanogaster* subgroup (183 of *D. melanogaster*, 88 of *D. simulans*, 43 of *D. sechellia*, 12 of *D. mauritiana*, 13 of *D. teissieri*, 36 of *D. yakuba*, 4 of *D. erecta* and 1 of *D. orena*), and of 1225 insect reference genomes from NCBI. It also presents results from the analysis of a sample of small-RNA sequencing data from ovaries of a *D. simulans* strain.

The invasion of *D. melanogaster* by Spink has already been published in the cited publication by Pianezza et al., 2023. In the latter publication, the presence of Spink in *D. sechellia* and *D. simulans* has also been shown. The presence in *D. willistoni* is implied since Spink's internal sequences are 99.7 % identical to the *D. willistoni* TE Gypsy-78_DWil. What's new here for Spink is its presence in *D. mauritiana* and the occurrence of related copies in drosophilid species outside of the *melanogaster* subgroup. The presence of Shellder in *D. simulans* and *D. mauritiana* has been published in the cited paper by Ding et al., 2016. A TE showing high similarity with Shellder, Gypsy-29_DWil, had been identified by others in *D. willistoni*. What is new in this manuscript is the extensive search for Shellder and Spink in other drosophilid species. While invasion of *Drosophila* species by 7 other TEs over the last 200 years has been reported (references in Pianezza et al., 2023), and 101 putative events of horizontal transfer have been proposed in *Drosophila* for 21 different elements (Loreto et al., 2008), the novelty here is that the authors suggest horizontal transfer and invasion by Shellder and Spink during the last 50 years.

Overall, this manuscript contains very interesting results and builds a nice story of recent species invasion by two transposable elements in *Drosophila*, it is of significance to the field. Meanwhile, the conclusions about horizontal transfer need to be reconsidered in depth, and

the title needs to be changed, replacing "horizontal transfers" with "invasions" (see my comments below).

Major comments:

Inferring horizontal transfer is quite tricky. The methods for inferring horizontal transfer are well described in the cited publication by Loreto et al. 2008. It is even more difficult to estimate when a putative horizontal transfer took place and from which donor species. The results presented in this manuscript are very interesting and there is a large amount of data, but the conclusions about horizontal transfer drawn from these results are not convincing. The statement in the title that "two retrotransposons triggered a cascade of horizontal transfers in *Drosophila* species within the last 50 years" is clearly overstated. Horizontal transfer between species over the last 50 years should create copies of the respective transposable elements that are almost identical to the donor element. This is only the case within the group of "invaded species" (Figure 4A), i.e. for Shellder in *D. simulans*, *D. teissieri*, and *D. mauritiana*, and for Spoink in *D. melanogaster*, *D. simulans*, *D. sechellia*, and *D. mauritiana*. In the other species, the "candidate donor species", the similarity is much lower. A good example of recent horizontal transfer, between 1950 and 1980, is the P element, which is 3 kb long and differs in just one nucleotide between *D. melanogaster* and *D. willistoni* (Daniels et al., 1990). Also, the Bayesian trees of Shellder and Spoink insertions in Figures 4B and 4C do not support very recent horizontal transfer in the last 50 years between one of the species of the melanogaster subgroup and a species not belonging to this group. This means that the horizontal transfer from the "candidate donor species" to the "invaded species" more likely is a quite ancient event and that the scenario presented in Figure 5C, and especially the dating of horizontal transfer events, is extremely unlikely. It can clearly not be stated that all the horizontal transfer events suggested here have occurred in the last 50 years. It is more appropriate to conclude that the donor species or population from which Shellder and Spoink invaded the melanogaster subgroup is not present in the 266 samples analysed here, as none of the "candidate donor species" in Figure 4 have copies of Shellder or Spoink that are nearly identical to copies of the "invaded species". A possibly functional TE related to Spoink, Gypsy-78_DWil, had been identified by others in *D. willistoni*. Gypsy-78_DWil shows 99.73% similarity with Spoink for internal sequences, but only 93.54% identity for LTRs (Pianezza et al., 2023). No possibly functional Shellder copy, having full coding capacity and identical LTRs, is reported for the putative donor species in this manuscript. Consequently, the donor species cannot be clearly identified and the time of horizontal transfer cannot be determined. Only eventual horizontal transfer events between the "invaded species" may be relatively recent, between species of the melanogaster subgroup, *D. melanogaster*, *D. simulans*, *D. sechellia*, *D. mauritiana* and *D. teissieri*. So the hypothesis of a recent "cascade of horizontal transfers" is not supported by the data. In addition, the events may not be independent, as shown in Figure 2, which suggests that Spoink and Shellder were transferred together. And finally, the divergence of the Shellder and Spoink copies in the different genomes (Figure 4B, C) rather suggests that there is a timelaps between the moment of horizontal transfer and the time of invasion. It's important not to confuse the two. It is more likely that the horizontal transfer between species occurred

some time ago already, more than 50 years ago, and that the invaded population only recently spread and 'contaminated' other populations of the same species around the world. The time of invasion is clearly not the time of horizontal transfer. This is mentioned in the manuscript: "... there could be a considerable lag time between the initial transmission of a TE and its spread in populations." (line 362) but not really considered when concluding for the time of horizontal transfer events. This should be considered all over the manuscript and discussed. The title needs to be changed, as only invasion has occurred recently with any certainty, and not horizontal transfer.

The TEs are highly similar between the species likely involved in HT: *Spoink* has a sequence identity of 99.59% between *D. willistoni* and *D. melanogaster* and *Shellder* 99.31% between *D. cardini* and *D. simulans*. In the revised manuscript we are also mentioning this high sequence similarity in the text:

"The sequence identity between the consensus sequence of *Spoink* and insertions for example in *D. willistoni* is very high (99.59% over 4662 bp)."

and

"The sequence identity between the consensus sequence of *Shellder* and insertions in these species is very high (e.g. *D. cardini* 99.31% over 6640 bp and *D. anceps* 99.29% over 6639 bp)."

Considering the greater nucleotide sequence variation of *Spoink* and *Shellder* compared to the *P*-element, we want to note that the *P*-element is a DNA transposon while *Shellder* and *Spoink* are LTR retrotransposons. That is, *Spoink* and *Shellder* have an RNA intermediate (unlike the *P*-element). Since reverse transcription is error-prone (PMID: 2460925) a somewhat higher nucleotide diversity is expected for *Spoink* and *Shellder* as compared to the *P*-element.

In the revised manuscript we added two supplementary figures (S14 and S15) that support the horizontal transfer of *Spoink* and *Sheller*. With this novel analysis we show that *Spoink* and *Shellder* fulfill the three classical lines of evidence for a HT: i) a patchy distribution of the TE among closely related species (Fig 4A) ii) discordant phylogenetic trees between host species and TEs (novel supplementary figure S14) and the iii) divergence of the TEs lower than the host genes (novel supplementary figure S15). With our time series data we are however providing even more compelling (i.e. direct) evidence showing the absence of these TEs in old strains and their presence in young ones.

The reviewer is correct in that we do not know if the HT happened within the last 50 years. Nevertheless we note that the HT cannot be older than 200 years, since no physical contact existed between the involved species prior to this time (e.g. *D. melanogaster* and *D. simulans* were absent from the Neotropics >200 years ago). This sets an upper limit for the time of the horizontal transfer of 200 years.

We however provide evidence that the TEs spread within the last 50 years. As suggested by the reviewer we thus changed the title to "Two retrotransposons triggered a cascade of invasions in *Drosophila* species within the last 50 years".

Moreover in the revised manuscript we are now discussing that the exact time of the HT is unknown, but cannot be much older than 200 years, i.e. the time of the habitat expansion of *D. melanogaster* and *D. simulans* into the Neotropics (the habitat of the donor species).

lines 23-25:

"Our work reveals that cascades of TE invasions, likely initiated by human-mediated habitat expansions, could have a profound impact on the genomic and phenotypic evolution of many geographically dispersed species."

This generalisation seems exaggerated, as these observations have only been made for drosophilid species and the mode of horizontal transfer is not known.

We weakened the statement to "Our work reveals that cascades of TE invasions, likely initiated by human-mediated habitat expansions, could have an impact on the genomic and phenotypic evolution of geographically dispersed species."

We however still think that this important message should be in the abstract. We discovered this phenomenon (cascade of TE invasions) in one of the prime model organisms for evolutionary and ecological research: *Drosophila*. The amazing set of genomic resources we could harness to identify the "cascade of invasions" (i.e. genomes of strains sampled during the last 200 years; genomes of many related species) is only available for very few organisms. Thus, it is feasible that cascades of invasions will be identified in other organisms once more data become available. Since we think that TE invasions could have a notable impact on genome and phenotypic evolution, we want to encourage future research to test whether such cascades of TE invasions can also be found in other species, outside of drosophilids.

line 122:

It would be important to note here that Gypsy-29_DWil is a TE that has been identified in *D. willistoni* and discuss this important point.

We added in brackets "identified in *D. willistoni*; ..." but we do not see why this merits discussion. *Shellder* is present in many species (esp. in the Neotropics). The fact that the *Shellder* version of *D. willistoni* rather than for example *D. cardini* (or any other species having *Shellder*) was present in RepBase reflects chance/human bias (i.e. researchers focusing on the highly abundant *D. willistoni*) more than biology.

line 124:

"We found that gypsy-29_DWil is identical (99.9%) to Shellder, a TE previously described by Ding et al. [2016]."

The authors indicate 99.9% identity of Shellder with Gypsy-29_DWil. This is not correct. Comparing Shellder, published by Ding et al., 2016, Genbank record KX196449.1, and Gypsy-29_DWil (from Repbase), there is only 97% identity between the internal sequences and 88% identity between the respective LTRs. If Gypsy-29_DWil were 99.9% identical to

Shellder from *D. simulans*, this would have been the only strong argument in favour of a recent horizontal transfer from *D. willistoni* to the melanogaster subgroup. But this is clearly not the case.

The reviewer is correct. The match with *Gypsy-29_DWil* and *Shellder* is indeed only 97%. However, the match between *Shellder* and the sequence in *D. sim* (SZ232) were *Gypys-29_Dwil* aligns is 99.9%. We changed the sentence to:

“When extracting the sequences of the *gypsy-29_DWil* hits in SZ232, we realized they were identical (99.9%) to *Shellder*, a TE previously described by Ding et al. [2016].”

We also note that *Shellder* has a higher similarity (>99.1%) with insertions found in other species (*D. leonis*, *D. anceps*) than with *D. willistoni* (97%).

line 141-143:

Although 88 lines have been analysed for *D. simulans*, many of them were collected at the same time in the same country, possibly even in the same locality: 19 of them were collected in Kenya in 2018, 11 in Kenya in 2006, 12 in Madagascar in 2002 etc., which means the sampling is in fact quite limited from a statistical point of view. Thus, the statement "This suggests that our method accurately captures the timing of TE invasions" (line 149) is clearly exaggerated.

We agree that more data, from more diverse spatial and temporal distributed populations, would be ideal.

Nevertheless a previous work (PMID:26982327), based on 631 *D. simulans* strains sampled around the world, showed that the *P*-element was rare among the *D. simulans* strains in 2006 but frequent in 2014. This is exactly what we find based on the 88 lines: the *P*-element is rare in 2006 (and before) but frequent in 2014 (and after). Although we agree that we do not have the same (spatial) resolution as PMID:26982327. Thus, we weakened the statement to: “This suggests that our approach, based on 88 sampled *D. sim* lines from different areas, allows us to capture the approximate timing of TE invasions.”

lines 151-153:

"The presence of *Shellder* and *Spoink* in one strain collected 1975 in Kenya (Ken75, SRR22548178) may be due to contamination of the strain."

This seems purely speculative and not based on any data, e.g. contaminated and non-contaminated samples from this Kenya 1975 strain. Then, how can the authors say that this strain is contaminated and not the others? It is entirely possible that the 1975 Kenyan sample came from a population in a different region of Kenya to the other samples. Did the authors check this? It is much more likely that the invasion had already begun in 1975. Importantly, the 1975 Kenyan sample is not shown in supplementary fig. S3 while the authors refer to this Figure (line 155). The corresponding results need to be shown and a non-contaminated sample of this 1975 Kenyan strain would be a good thing to find.

Contamination of strains can happen. For example, a previous work found that 1 out of the tested 40 DGRP strains was contaminated

<https://genomebiology.biomedcentral.com/articles/10.1186/s13059-019-1912-z>. We also note that for TEs, contamination has a direction, i.e. a contamination can only lead to the 'false' presence of a TE never to the 'false' absence. Since TEs amplify, a single fly carrying a novel TE may be sufficient to contaminate a fly strain. In other words, the absence of a TE cannot be due to contamination, but the presence can. Compared to other strains collected around 1975, the strain Ken75 is a single outlier having both *Spoink* and *Shellder*. We thus wanted to avoid overinterpreting a single outlier, especially given that contamination of strains can happen. But it is, of course, also feasible that the invasion had already begun in 1975.

Fig S3 shows the coverage for *Shellder* and *Spoink* in 6 (out of 88) example strains. Its purpose is to show that the coverage along the TEs is uniformly elevated in the 3 strains having the TE, compared to strains sampled before the TEs invaded. Its purpose is not to infer the exact timing of the invasion. To infer the timing of the invasion we show Fig 1 in the main manuscript based on all 88 strains. Ken75 would just be an alternative example for the resulting coverage when *Spoink* and *Shellder* are present.

Unfortunately, we used all the available resources we could find in this work and we did not find another strain from Kenya sampled around this time.

However, to address the issue that the invasion may have already started in 1975, we changed the manuscript to the following.

“The presence of *Shellder* and *Spoink* in one strain collected 1975 in Kenya (Ken75, SRR22548178) is possibly due to contamination (although it is feasible that the invasion of both TEs already began in 1975).”

Although many genomes of different species have been analysed, the data are not always sufficient to justify the conclusions drawn. It would be enough to find a single *Drosophila* somewhere, among museum specimens for example, from before 1974 and having *Shellder* and/or *Spoink* to invalidate the hypothesis of horizontal transfer and invasion only after 1974. It is impossible to determine when horizontal transfer took place, as there are not enough genomic sequencing data for ancient times. It is therefore clearly an overstatement to claim that horizontal transfer and invasion have occurred in the last 50 years, and not before. Nevertheless, the results presented strongly suggest that the invasion of the studied populations is recent. An interesting question is then to determine which population was the first to be contaminated and where it came from. The authors attempt to answer this question, but here too we come up against a lack of available samples: the 88 genomes analysed for *D. simulans* in Supplementary Table 2, for example, come from only 22 different regions; only 5 genomes come from *D. simulans* captured before 1974, and of these, 2 correspond to the same strain w501 (Figure 1, supplementary Table 2). The same is also true for the other species, where many samples originate from the same regions (see suppl. Tables 7-10). Thus, we cannot be sure that *Shellder* and *Spoink* were absent from a given species before 1974.

We fully agree that it is impossible to estimate when and where a horizontal transfer actually happened. Identifying the first individual that carried a novel TE would require sequencing every fly that ever lived, which is clearly not feasible. However, in our case we can narrow

down the timing to 200 years. Our phylogenetic analysis indicates that the donor of *Spoink* and *Shellder* are species from South America.

Both recipient species, *D. simulans* and *D. melanogaster*, solely extended their habitat into the Americas about 200 years ago. Hence, the HT cannot be older than 200 years. We think that we have good evidence that the invasion occurred during the last 50 years. For example, fig 1 shows that *Spoink* and *Shellder* are largely absent in the analyzed strains before 1994, they are present in most strains between 1994 and 2016, and they are present in all strains following 2016.

We are now discussing the timing of the horizontal transfer in more detail:

“Furthermore, we note that the time of the horizontal transfer may have occurred earlier, since there could be a lag-time between the transmission and the spread in natural populations. We however also note that the horizontal transfer of both TEs, could not be much older than 200 years since *D. simulans* and *D. melanogaster* were absent in South America (i.e. the habitat of the putative donor species of *Spoink* and *Shellder*) before that time (Sturtevant1921,Johnson1913,Keller2007).

Shpak et al., 2023 sequenced 25 genomes from museum specimens of *Drosophila melanogaster* ranging from the early 1800s to 1933 ("Genomes from historical *Drosophila melanogaster* specimens illuminate adaptive and demographic changes across more than 200 years of evolution", PLoS Biol 21: e3002333). As I could not find a file with the analysed short-read genomic data sets, I could not see all the strains that were used in this study. Such a list should be joined to the supplementary. Did the authors study the genomic data from Shpak et al. 2023? This publication is not listed in the references.

We only considered one sample from 1933 (Shpak et al 2023) in the analysis since we wanted to focus on the relevant time-window in Fig S7. The *Spoink* invasion happened around 1993. Showing strains collected around 1800 would thus only lower the resolution for the relevant time window (i.e. 1993±50years). This single sample from Shpak in 2023 is mentioned in Supplementary Table 2 (overview of all analyzed *D. mel* data). However, we analyzed the 25 samples published by Shpak et al 2023, with the same approach used in the manuscript. *Spoink* and *Shellder* are absent from all samples (several of the 25 strains are overlapping in the figure)

lines 159-161:

"... worldwide populations prior to 1994 ..."

The term "worldwide populations" suggests that many different populations captured at many different places "worldwide", all over the world, were used in this study. This is not the case. There are only 13 genomes from *D. simulans* captured prior to 1994 that have been analysed and these originate from 7 to 9 different regions (supplementary table 2).

We have changed this as follows: "When investigating the geographic spread, we found that both TEs were absent in strains sampled prior to 1994"

Figure 4:

There is some incongruence between Figures 4A, 4B, 4C and the list of analysed data sets in supplementary file S1. For example, there are only 7 data sets for *D. melanogaster* in supplementary file S1 while there are 9 in Figure 4A and 17 data sets for *D. melanogaster* in Figure 4C. To be able to fully comprehend these data, the source data in a supplementary table would be required. It would also be necessary to give the results and references for all analysed data sets.

There are actually 9 entries for *D. melanogaster* in Supplementary file S1. In some cases the species name was abbreviated. To make this more clear we added a novel column to supplementary file S1 with the full species name. These are the genomes we used for figure 4A.

In Fig 4C we are **not** showing 9 assemblies, we are showing the 17 insertions of *Spoink* in a **single** assembly. We have clarified this in the description as follows: "Multiple entries for a single species represent different insertions of a TE family in a single assembly."

Supplementary file S1:

35 of the 266 data sets in supplementary file S1 are not yet publicly available (annotated "PENDING" and "xxx ... Sarah"), and 29 of these data sets come from other authors. Will these source data be made accessible and when? It would also be necessary to indicate the full species name for each data set.

All data sets have now been made publicly available. We revised supplementary file S1

- we added the GCA number for all of the assemblies which have been made publicly available after our submission,

- we also fixed the mistakes mentioned by the reviewer

- we added a new column with species and genus names (see above).

- The *D. mauritiana* genomes are now publicly available and the GCA numbers have been added to our file. Here is a screenshot of NCBI:

[ ]	ASM3965464v1	GCA_039654645.1	Drosophila mauritiana	Dmau_R31 (strain)	⋮
[ ]	ASM3965466v1	GCA_039654665.1	Drosophila mauritiana	Dmau_R61 (strain)	⋮
[ ]	ASM3965470v1	GCA_039654705.1	Drosophila mauritiana	Dmau_R32 (strain)	⋮
[ ]	ASM3965465v1	GCA_039654655.1	Drosophila mauritiana	Dmau_R39 (strain)	⋮

line 323:

"... we found insertions with a high similarity to Spoink and the P element in species of the willistoni group (fig. 4A; supplementary fig. S12)."

The term "high similarity" is exaggerated and not precise, unless there are other data that are not shown. The similarity seems to be less than 97%, which is not "high" for a hypothetical horizontal transfer less than 50 years ago. The similarity between Spoink and Gypsy-78_DWil from *D. willistoni*, high for internal sequences, relatively low for LTRs (see above, Pianezza et al., 2023), might be discussed here.

The "similarity" displayed on the y-axis in figure 4A is NOT the raw sequence identity in percent, but rather the proportion of the maximum Smith Waterman alignment score (as provided by RepeatMasker) in a genome versus the maximum score across all the assemblies (see Material and Methods). We think that this is an elegant approach that allows us to capture the goodness of a match (between TE and an assembly) with a single key-metric (instead of two, i.e. raw sequence similarity and length of the match; for more details please see our response to reviewer 2). An exhaustive description would be unwieldy for readers, especially in the results section. Therefore we were searching for a short descriptive term that captures the essence of the idea and felt that 'similarity' is the most suitable analogy.

To clarify we are now writing "...high similarity (based on SW score; see M&M).." Additionally we changed the y-axis of Fig 4A to "similarity (SW score)".

To clarify that the actual sequence similarity is much higher we also added:

"The sequence identity between the consensus sequence of Spoink and insertions for example in *D. willistoni* is very high (99.59% over 4662 bp)."

and

The sequence similarity between the consensus sequence of *Spoink* and other species of the willistoni group (*D. willistoni*, *D. paulistorum*, *D. tropicalis*, *D. equinoxialis*) is actually >99% (see for example the following link: here).

line 327:

"... insertions with a high similarity to Shellder were found in species of the cardini and the repleta groups"

As before, the term "high similarity" is exaggerated and not precise. It seems to be less than 97 or 98%, which is not "high" for a hypothetical horizontal transfer less than 50 years ago.

This comment is similar to the one above. This similarity measure is based on the Smith-Waterman alignment score. The raw sequence similarity between the consensus sequence of Shellder and insertions in *D. cardini* and *D. parthenogenetica*, *D. anceps*, *D. leonis* and *D. pegasa* is actually higher than 99%. To make it more clear that our similarity metric is based on the Smith-Waterman alignment score we are now writing: "...similarity (based on SW score; see M&M).

To clarify that the actual sequence similarity is higher we added the following:

"The sequence similarity between the consensus sequence of *Shellder* and insertions in these species is very high (e.g. *D. cardini* 99.31% over 6640 bp and *D. anceps* 99.29% over 6639 bp).

line 338:

"Spoink insertions of the melanogaster subgroup are nested in insertions from the willistoni group (fig. 4B)"

What is meant by "nested" here? Spoink copies of the willistoni group appear to be on distinct branches to copies of the melanogaster subgroup. Please clarify this important point.

"fig. 4B" should be replaced with "fig. 4C" for Spoink.

We have corrected the reference to 4B and replaced it with 4C. By 'nested' we mean that the insertions (red) are phylogenetically placed within the branch that contains species from the willistoni group (light blue). This is clearly discordant from the phylogeny of the species (see novel supplementary Fig. S14). It is perhaps difficult to see because the branch lengths are very short (i.e. little divergence between groups). See below a small mockup of what we mean:

To clarify this point in the text we have edited as follows: “*Spoink* insertions of the melanogaster subgroup are nested within the willistoni subgroup (i.e. the branches from the melanogaster group are contained within species of the willistoni group; fig. 4C), in agreement with our previous work suggesting that a horizontal transfer from a species of the willistoni group triggered the recent *Spoink* invasion (Pianezza 2023). “

In addition to figure 4, it would be useful to include a table with the species in each group (repleta, cardini, saltans, willistoni, melanogaster) and the source data from figure 4A in a table.

We made available a table with all of the assemblies used for each of the species groups mentioned by the reviewer:

<https://github.com/rpianezza/DoubleTrouble/blob/main/1-REVISION/species-groups.tsv>.

Moreover, we now also provide a table with the raw data used for figure 4A (<https://github.com/rpianezza/DoubleTrouble/blob/main/1-REVISION/fig4A-rawdata.tsv>) and more statistics (e.g. genomic coordinates, sequence identity, alignment length, etc).

In the revised manuscript this table is additionally provided as novel Supplementary File S3.

line 341:

"Shellder insertions of the melanogaster subgroup are nested within species of the cardini and repleta groups, suggesting ..."

As before, what is meant by "nested"? Shellder copies of the repleta and cardini groups appear to be on distinct branches to copies of the melanogaster subgroup. Please clarify this important point.

See above. To clarify we added an explanation of the term “nested” in the *Spoink* paragraph (i.e. the branches from the melanogaster group are contained within species of the willistoni group; fig. 4C).

It would be useful to give the shortest phylogenetic distances between copies of putative donor groups and the invaded groups and compare these to the longest phylogenetic distances within each group. The data in Figures 4B and 4C alone, do not support recent horizontal transfer, less than 50 years ago, between the "candidate donor species" and the "invaded species".

We are not entirely sure if we correctly understand this comment (with shortest and longest phylogenetic distance). But to support the recent HT we added two novel supplementary figures that show the discordance between the phylogeny of the host species and the TE (supplementary Fig S14) and the divergence of the TEs compared to the divergence of the host genes (Supplementary Fig. S15). Both novel figures strengthen our claim for the recent HT. Furthermore we added novel paragraphs, stating that the sequence similarity between the consensus sequence and the putative donor species is very high (>99%; see comments above; line 323, line 327) which supports the recent HT.

Line 435:

Loreto et al., 2008, should also be cited for the three methods.

We are now citing Loreto et al 2008 throughout the manuscript in support of the statement that "... horizontal transfer of a TE is inferred from three lines of evidence: i) a patchy distribution of the TE among closely related species, ii) a phylogenetic discrepancy between the TE and the host species and iii) a high similarity between the TE of the donor and recipient species, which is frequently quantified by the synonymous divergence of the TE [Peccoud et al., 2017, Wallau et al., 2014, Loreto et al., 2008]."

lines 438-444:

These conclusions about presence/absence of Shellder and Spink in the respective species should be tuned down because of the relatively low number of ancient samples.

We tuned down the conclusions:

"... we showed that Spink insertions were most **likely** absent in worldwide *D. melanogaster* populations before 1983 but present in all populations after 1993.

Based on similarly compelling evidence from time-series data, we show in this work that Shellder and Spink were **largely** absent from the **investigated** *D. simulans* strains sampled before 1995 but present after 2012...."

lines 471-484:

It is not clear why the authors do not consider the possibility of a joint transfer of both Spink and Shellder from the same donor for transfer within the melanogaster subgroup, eventually coupled to high copy number for both TEs and transposition of both TEs in the invaded populations, increasing their copy numbers. Please clarify.

Yes, the reviewer is correct. We added the following to the discussion"

Another option is that both Shellder and Spoink got transmitted to *D. simulans* from the same donor species. Our results suggest that Shellder and Spoink, are derived from different species (Spoink from *willistoni* and Shellder from the *cardini* group) making this hypothesis unlikely. However, we cannot rule out that a species that has not yet been sequenced acted as common donor of both TEs.”

Minor comments:

Abstract, line 18:

The authors propose hybridisation between species as the mode of horizontal transfer in the abstract. But hybridisation is only one of the multiple hypotheses that could explain horizontal transfers (see the cited Loreto et al., 2008). There is no strong argument in favour of species hybridisation. "Likely" should be replaced with "Possibly".

Fixed.

line 35:

TEs cannot be compared to extracellular disease-causing pathogens simply by describing them as "intragenomic parasites". TEs are not viruses. Please tune down.

Edited as follows: "Intragenomic invaders, such as transposable elements (TEs), may not be exempt from these dynamics - as species ranges change and humans transport organisms around the planet, they will come into contact with novel genomic invaders such as TEs. "

line 61: Also cite Desset et al., 2003, for flamenco here, as "COM" and "flamenco" correspond to the same piRNA cluster.

We have added this citation.

line 78:

"Given the small number of known TE families in *D. melanogaster* (124) and the high number of TE invasions in the last 200 years, it is unlikely that this rate of invasions is the normal 'background' level of introduction.

The rate of invasion cannot be estimated solely on the basis of the number of known TE families.

There are at least 180 known TE families in *D. melanogaster* in the databases (Dfam, Repbase). Please check. In addition, some TEs have been published but do not appear in TE databases (like Spoink and Shellder). Furthermore, to date, only a few strains of *D. melanogaster* have been extensively studied for their TE content and the number of *D. melanogaster* genomes from around the world remains relatively small. Therefore there are probably more TEs to be discovered. The *D. melanogaster* genome is also full of dead transposable elements that invaded the species in the past and we do not know how long

they may remain active. It is therefore possible that exchanges between populations and subsequent invasions are frequent in *Drosophila*. More data, particularly from old museum specimens, are certainly needed to estimate the rate of invasions at different periods in the history of *Drosophila*.

We agree that a robust inference of the rate of invasion requires more data of ancient *Drosophila* specimens. Ideal would be specimens dating back several thousands of years (e.g. we are exploring some options). In the absence of these data we are limited to the current state of the field unfortunately. The most extensive recent scan of *D. melanogaster* genomes comes from the lab of Josefa Gonzalez (PMID:35413957). This is the library that included a partial sample of *Spoink*. The number of TE families uncovered (165) is similar to the TEs present in the set of consensus sequences maintained by Casey Bergman (127). So while one may dispute the exact number of TE families, it will not change the overall conclusion that this number of recent invasions is unexpected given the number of existing families of TEs in *D. melanogaster*. We have changed our discussion to make it clear that the number of existing families is an estimate, thus there is an amount of uncertainty to our claims: "Given the **estimated** number of known TE families in *D. melanogaster* (124+) and the high number of TE invasions in the last 200 years, it is unlikely that this rate of invasions is the normal 'background' level of introduction."

line 122:

"... second TE annotated as gypsy-29_DWil (from the repeat library of [Chakraborty et al., 2021]) ..."

The citation is not quite correct. Chakraborty et al., 2021 did not identify or describe any *D. willistoni* TEs. The "Gypsy-29_DWil" TE has been identified in *D. willistoni* and submitted to Repbase records by Jurka J. and Kohany O. in 2011. The publication cited here should refer to Repbase: Repbase, <https://www.girinst.org/>, Jurka J, Kapitonov VV, Pavlicek A, Klonowski P, Kohany O, Walichiewicz J. Repbase Update, a database of eukaryotic repetitive elements. Cytogenet Genome Res. 2005;110(1-4):462-7. doi: 10.1159/000084979. PMID: 16093699.

Fixed.

line 133:

"Spoink belongs to the gypsy/mdg3 family (members do not have env ..."

TEs that do not have an env are not necessarily of the same clade and I have no knowledge of a "gypsy/mdg3 family". Please check the names of the clades. The Micropia/mdg3 clade has been described by Malik and Eickbush, 1999, J.Virol..73:5186-5190 (see also <https://gydb.org/index.php/Ty3/Gypsy>).

In an important study Kapitanov and Jurka 2003 PNAS (<https://www.pnas.org/doi/full/10.1073/pnas.0732024100>) classify the LTR transposons, and introduce the different families, including the 'GYPSY/mdg3' family in Table 3. In this Table 3 they also show that members of the gypsy/mdg3 group do not encode an 'env'. We are citing this work in support of our claim.

line 134:

"In contrast to *Spoink*, *Shellder* may thus generate virus-like particles that could infect other species or tissues (e.g. the germline)."

The authors suggest here that *Shellder* could be a retrovirus. Until now, infectious virus-like particles involved in horizontal transfer between species have been suggested but there is no clear evidence for this mode of horizontal transfer in *Drosophila*. This should be clarified.

Edited as follows: In contrast to *Spoink*, *Shellder* may generate virus-like particles that could infect other species or tissues (e.g. the germline), though this type of activity is at present largely hypothetical.

lines 136-137:

"*Shellder* and *Spoink* are both biased towards intergenic insertions, though *Spoink* inserts into introns more frequently than *Shellder* (30% versus 20%). Neither TE frequently inserts into other genic features, in contrast to the P-element where 42% of insertions are in promoters (supplementary fig S2)."

This statement may be incorrect as the data in supplementary fig S2 were apparently not normalized to the global size (or genomic fraction) of the respective regions in the genome. For example, promoters represent a very small fraction of the genome. Therefore, the probability of insertion into promoters is rather low compared to other regions. Idem for the legend of fig S2 where it is said that "*Shellder* has a pronounced insertion bias into intergenic regions."

Our statements refer only to the proportion of insertions in a genic feature, with no statements about the portion of the genome that is occupied by a given type of feature. For example, when we say that 42% of insertions of the P-element are into promoters, this is out of the total number of insertions. If we were to calculate the size of the genome, the size occupied by any given feature (promoter, 3' UTR, 5' UTR), and therefore the likelihood of inserting into each type based on the amount of the genome occupied, that would be a very different calculation, that requires reliable genome annotations. Given that we are working with non-model genomes this number would be quite approximate. We thus think that the % of insertions serves to illustrate the point about insertion bias. Also comparisons among TEs based on the same approach are likely valid. However, we aimed to make this clearer as follows: *Shellder* and *Spoink* are both biased towards intergenic insertions, though *Spoink* inserts into introns more frequently than *Shellder* (30% of insertions versus 20%).

The data in supplementary fig S2 concern only *D. simulans*. This must be specified.

The revised figure caption states: Summary of insertion locations for *Shellder*, *Spoink*, and the P-element in *D. simulans*.

Furthermore we added in the main manuscript "... (supplementary fig. 2; **based on *D. simulans***).

Figure S2 concerns results from 10 (for Shellder and Spoink) or 3 genomes (for the P-element). The standard deviation between genomes should be shown in the figure.

We have added the standard deviation to this figure.

What is meant by "other genic features" ? Please specify.

We edited the text to: "...into other genic features (i.e. 5' UTR, 3' UTR, exon, promoter),...".

line 183:

"If Shellder is a somatic piRNA cluster ..."

Shellder is not a piRNA cluster but a transposable element.

Fixed. "If Shellder is a somatic TE..."

line 199:

"Furthermore, Shellder and Spoink are controlled by the piRNA pathway in *D. simulans*."

There is no clear evidence in the manuscript that Spoink is controlled by the piRNA pathway in *D. simulans*.

We changed the sentence as follows, providing the reference to the Supplementary figure where we show the piRNAs response in *D. simulans*.

"Furthermore, Shellder and Spoink seem to be controlled by the piRNA pathway in *D. simulans* (supplementary fig. 5)"

line 202:

"Shellder is absent in *D. melanogaster*"

Even if 183 genomes, corresponding to 183 laboratory strains or wild-caught strains, often derived from a single female, have been analyzed here, it is still possible that other *D. melanogaster* strains or populations exist that have Shellder. In addition, there are only 2 very recent strains, collected in 2015 and 2018, among the 183 strains. "Shellder was not found in *D. melanogaster*" would be a better title for this section.

We changed the title of the section in "*Shellder was not found in D. melanogaster*", as the reviewer suggested.

The same considerations also apply to the "Shellder, but not Spoink, invaded *D. teissieri*" section (line 263), for example, where 15 strains were analyzed, only 8 of which were certainly recent (the Bioko strains).

We changed the title to "*Shellder, but not Spoink, was found in D. teissieri*"

Ditto, "Shellder and Spoink are absent in other species of the melanogaster subgroup" (line 292) should be reworded to "Shellder and Spoink have not been found in other species of the melanogaster subgroup".

Edited: "Shellder and Spoink were not found in other species of the melanogaster subgroup"

Figure 4, legend:

"full-length insertions (> 80% of length)"

80% is not "full-length", only 100% can be considered as "full-length". LTRs are often found different between related TEs. Do the authors mean full-length, i.e. 100% of length, alignment for the internal sequences only? Please specify or reword to avoid the term "full-length". The same goes for other figure legends using the same terms.

The description was not complete: we indeed use a threshold of 80% of the length to include a TE insertion into the tree, but the insertion also has to contain the two LTRs of *Shellder* at the ends of the sequence (found by BLASTn). We added this information. Additionally we changed 'full-length insertion' to 'complete insertion'. Finally in response to a comment of Reviewer 2, we moved this description to the M&M.

The legend of Figure 4B refers to both 4B and 4C.

Fixed

Replace "266 *Drosophila* species" with "266 drosophilid species" and verify whether all 266 drosophilid species or the 209 *Drosophila* species are shown in Figure 4A.

We changed the sentence to "266 long-read assemblies representing 243 drosophilid species", as it is more precise. All of the 243 species and 266 assemblies are shown in figure 4A.

Supplementary Figure 8:

"... hits with less than 750bp length or more than 10% divergence were removed..."

Does this mean that a fragment of e.g. 1kb length and having 9% divergence from the respective TE was considered as a "copy" here? In this case, it would be suitable to show also the copy number of full-length TEs with less than 2% divergence from the reference element for example to have an idea of TEs that potentially transposed less than 50 years ago. Indeed, ancestral copies of *Drosophila* TEs, which are not functional anymore (mutated ORFs) often show between 5 and 10% divergence, or even less, with the functional TE. Such ancestral copies are mostly estimated being older than 50 years. The literature about the rate of mutations affecting TEs in *Drosophila* should be studied and considered here.

We used the 750bp threshold solely because of the P-element, where many recent strains only harbor small internally deleted copies of the P-element (see for example PMID:33210145,2832152).

The figure below has been made with the thresholds suggested by the reviewer (2% divergence, 5200bp for *Spoink* and 2900bp for the P-element). The results are qualitatively unchanged for *Spoink* and *Shellder* (copy number estimates changed for *Spoink* because of truncated copies; for examples of truncated *Spoink* insertions see Fig 1B in <https://www.biorxiv.org/content/10.1101/2024.04.25.591091v1.full.pdf>).

Supplementary Figure 13, legend:

"... in 1226 long-read assemblies of insect species"

Are the authors sure that all 1226 assemblies are long-read assemblies, i.e. from Nanopore or PacBio sequencing ?

The source data should be made available to identify the species shown in figure S13A.

We changed the sentence: "... in 1226 reference genomes of arthropods species", as the assemblies were not all based on long-reads.

The source data used to generate figure S13A has been uploaded on GitHub (<https://github.com/rpianezza/DoubleTrouble/blob/main/1-REVISION/arthropods-similarity.tsv>).

line 297:

"... infected by Shellder ..."

Shellder is not a virus. The term "invaded by Shellder" should be preferred.

Fixed

line 300:

"were also absent from other species" should be replaced with "were not found in other species", given the low number of samples analysed for these species.

Fixed: "Similarly *Shellder* and *Spoink* were not found in other species of the melanogaster subgroup"

line 307:

"We thus concluded that Shellder and Spoink did not yet invade *D. yakuba*, *D. santomea*, *D. erecta*, and *D. orena*."

This conclusion is overstated given the low number of samples analysed for these species and identical geographic origin for many of them (e.g. 29 samples for *D. yakuba* originate from Sao Tomé).

Restated as follows: The most likely conclusion is that *Shellder* and *Spoink* did not yet invade *D. yakuba*, *D. santomea*, *D. erecta*, and *D. orena*, though additional sampling is needed.

Supplementary Table S9:

Replace "36 analysed *D. yakuba* strains" with "38 analysed *D. yakuba* strains".

Fixed (Supplementary Table S10).

line 464:

"multiple species of the melanogaster subgroup" refers to 3 or 4 species, replace "multiple" with "several" for example.

Restated as follows: The invasions of Shellder and Spoink in several species of the melanogaster subgroup were likely triggered by horizontal transfer.

Reviewer #2 (Remarks to the Author):

Scarpa et al. presented a work that examined a recent spread of retrotransposons in *Drosophila* species.

They focused on two gypsy LTR retrotransposons, Shellder and Spink and examined the copy number in several *Drosophila* species, in particular, species from the melanogaster group, for which strains that were isolated across many years from different locations have been sequenced.

Authors found that both Shellder and Spink were found only in some, but in all assemblies of the same species, and from that concluded that this is evidence of a recent invasion.

major comments

I appreciate that authors examined in-depth the copy number, sequence divergence, and geographical and chronological spread of these elements in several species.

The sporadic presence of a particular TE can indicate either a recent introduction of the TE into a subpopulation of the species (source of which could be a horizontal transfer), or a recent activity of a TE insertion that already existed within the species.

The paper entirely focused on the possibility of the former. However the latter scenario could also create a similar distribution pattern. The paper should investigate or at least discuss this alternative scenario.

The reviewer suggests an alternative scenario of vertical inheritance. We think that this hypothesis has several problems.

- First, this hypothesis requires that the TE was consistently present in a few individuals of the species, i.e. Spink in *D. melanogaster*, *D. simulans*, *D. mauritiana*, and Shellder in *D. simulans*, *D. mauritiana*, *D. sechelia*, *D. teissieri*. However we did not find these two TE in any of the early collected strains (but found the TEs in all late collected strains).

- Second, it requires a mechanism that triggers the spread of the TE from a few isolated individuals to the bulk of the populations of several species in the last decades. We cannot think of any mechanism that could explain such a massive amplification in multiple species concurrently.

- Third, in a novel supplementary figure (S14) we show that the phylogeny of the TE is highly discordant with the species tree based on the BUSCO genes. This scenario is not expected with vertical transmission.

- Fourth, in a novel supplementary figure (S15) we show that the sequence divergence of the TEs is much lower than the divergence of any host gene for the pairs of species that were likely involved in the HT. It is not plausible that the sequence of a TE is more conserved than all of the investigated host genes.

In the revised manuscript we are now discussing the vertical transmission hypothesis proposed by the reviewer. Furthermore we added two novel supplementary figures (see below) showing i) the discordance between the species phylogeny and the phylogeny of the TEs and ii) the divergence of the TE and the host genes for the species likely involved in the HT

The sequence divergence of these elements, for example, of *Shellder* as shown in Figure 4B is largely congruent with the phylogeny of species, which implies that the element has been vertically inherited in a large picture.

We do not agree with the reviewer. The phylogenetic tree of the species and the TEs are highly discordant. To make this more clear we added a novel supplementary figure (S14) that shows the phylogenetic tree of species having full-length insertions of either *Spoink* or *Shellder*. We show a tree based on the host genes and a tree based on the sequence of *Spoink* or *Shellder*. The tree based on the host genes shows the expected clustering by species group whereas the tree based on the TE sequence shows a discordant intermixing of species from different groups. Such discrepancies in the phylogeny are usually considered as evidence supporting HT of the TE (e.g. PMID:29283188,22798449)

In another novel supplementary figure (S15), we compare the divergence of the TEs to that of the host genes for species pairs likely involved in the horizontal transfer of *Spoink* or *Shellder*. For both *Spoink* and *Shellder*, the divergence is significantly lower than that of any

host gene, providing strong evidence for horizontal transfer. Since TEs are expected to evolve under neutrality, the fact that *Shellder* and *Spoink* show less divergence than any host gene is inconsistent with vertical inheritance (especially when the TE was only present in small subpopulations, where the effect of drift should be magnified). A low divergence of the TE relative to host gene is typically considered good evidence for a recent horizontal transfer (see also PMID:29283188,22798449).

Authors focused on the connection between elements from the repleta group or cardini group that are closely related to elements found in the melanogaster group species.

This is not easily captured from the text nor from the figures.

RepeatMasker revealed that *Spoink* and *Shellder* insertions in *D. simulans*/*D. melanogaster* are most similar to insertions in the cardini, repleta and willistoni group. This is shown in Fig 4A , where the height of the bar (y-axis) indicates the similarity (based on Smith-Waterman alignment score) between the consensus sequence of the TE and the best matching sequence in the genomes of many different species (x-axis). For both *Spoink* and *Shellder*, the highest bars can be found with species from South America (such as repleta, cardini and willistoni). Hence these species from South America have *Spoink* and *Shellder* insertions that most closely resemble the consensus sequence from the melanogaster group. In Fig 4A

we highlight these species groups by assigning distinct colors to them and by mentioning solely the names of these highlighted species groups on top of the figure.

This is also mentioned in the text

“...we found insertions with a high similarity ($\geq 99\%$) to *Spoink* and the *P*-element in species of the willistoni group (Fig 4A..)”

and further down

“On the other hand, insertions with a high similarity to *Shellder* were found in species of the cardini and the repleta groups, and to a lesser extent in species of the willistoni and saltans groups (fig. 4A)”.

To further highlight these species in the revised manuscript we added the colors of the species group in the text, eg.: ...we found insertions with a high similarity ($\geq 99\%$) to *Spoink* and the *P*-element in species of the willistoni group (Fig 4A; **turquoise shades**)”

I initially misread that the horizontal transfer occurred from a willistoni group species to a melanogaster group.

But this is hardly possible given that the similarities between the willistoni *Shellder* elements and all the melanogaster *Shellder* elements are generally low ($\sim 80\%$ according to Figure 4B).

The *Shellder* invasion in *D. simulans* was likely triggered by a HT from a species of the cardini or repleta group (not from willistoni).

In Figure 4B we show the phylogenetic tree based on the similarity of the TE sequence. The extent of the actual sequence similarity cannot be seen nor inferred from the figure, we are therefore surprised that the reviewer reports a sequence similarity. In the caption of Fig. 4B we mention that we used solely full length insertions, which we defined to have at least 80% of the sequence length shared with the consensus sequence. Perhaps this arbitrary length cutoff of 80% is the reason why the reviewer assumed a sequence similarity of 80% between species.

However the sequence similarity of *Shellder* insertions between *D. simulans* and *D. cardini* is about 99%, (i.e. much higher than 80%).

To prevent this confusion in the revised manuscript, we removed the technical detail of the 80% length threshold from the figure legend and refer to M&M instead.

Additionally we are now more clearly mentioning that the *Shellder* invasion was triggered by HT from a species of cardini or repleta groups:

“This suggests that a horizontal transfer from a species of the cardini and repleta group triggered the recent *Shellder* invasions.”

Finally in the revised manuscript we are now mentioning the actual sequence similarity of *Shellder* between *D. simulans* and putative donor species:

“The sequence identity between the consensus sequence of *Shellder* and insertions in these species is very high (e.g. *D. cardini* 99.31% over 6640 bp and *D. aniceps* 99.29% over 6639 bp).”

I thank authors for putting details of genome assemblies used for the analyses and individual insertions in the supplemental tables.

By manually downloading the genomic sequences and clustering the open reading frames (ORFs), I could confirm that *Drosophila Mauritiana* R31 (GCA_039654645.1_ASM3965464v1) carries multiple copies of *Shell*der that are highly similar at the amino acid sequence level to the *Shell*der elements in *Drosophila anceps* (GCA_035045945.1_ASM3504594v1).

Given that the melanogaster group and the repleta group separated more than 30 million years ago, this is a strong evidence for a horizontal transfer event.

However, I did not see intact copies of *Shell*der in the *Drosophila similans* genomes, such as, the sz232 and MD242 strains, which authors claimed carry multiple insertions.

These include their top hits "tig00000792 pilon pilon 450324 456077 +" and "tig00000776 pilon pilon 2683380 2689137 C" in the sz232 strain (supplemental table 1).

This is misleading because fragmented copies decay rapidly and affect tracking of the evolution of these elements between species.

Please provide information of where the intact copies of each assembly can be found.

We thank the reviewer for the effort of reproducing our results. We are glad that the reviewer found insertions in R31.

Given the concern of the absence of *Shell*der in some *D. similans* assemblies, we repeated the analysis. We downloaded the two *D. similans* genomes from NCBI

- sz232 https://www.ncbi.nlm.nih.gov/datasets/genome/GCA_039725755.1/

- md242 https://www.ncbi.nlm.nih.gov/datasets/genome/GCA_039725595.1/

and performed a BLASTn search using the consensus sequence of *Shell*der

<https://github.com/rpianezza/DoubleTrouble/blob/main/sequences/Shellder-consensus.fasta>

We found multiple full-length matches of *Shell*der in MD242 (23 hits) and sz232 (6 hits). We provide the coordinates of the BLAST-hits in bed files and make them available to the reviewer:

<https://github.com/rpianezza/DoubleTrouble/tree/main/1-REVISION/Shellder-insertions>

The coordinates also allow extracting the sequences of BLAST-hits (e.g. with bedtools). We also make the sequences of the BLAST-hits available to the reviewer in fasta-files (in the same folder). These extracted BLAST-hits have both LTRs, a length of around 6600bp and a high similarity to the consensus sequence of *Shell*der (98-99%; as determined with a pairwise BLAST).

Additionally we updated Supplementary table 1 (and 5 and 7) with the novel contig names assigned by NCBI (we published the SZ232 assembly in a previous work; in the first submission we used our contig, i.e. the names of the contigs before they were renamed by NCBI; we apologize for this).

Along the same line, by simple blast search, I did not find any copies of *Shellder* with all three ORFs in *dsim_28_F0* (GCA_029774875.1_ASM2977487v1) or *dsim_29_F0* (GCA_029774795.1_ASM2977479v1).

Please provide more detailed information that formed the basis of supplementary Figure 4.

This is actually a very interesting case. The reviewer is correct, these assemblies do indeed not contain *Shellder* (nor *Spoink*). Interestingly, however, the short read data of these strains (ERR9453536, ERR9439680) do contain *Shellder* (and *Spoink*). Mapping the short reads to the consensus sequence of *Shellder* produced a clear and contiguous coverage along the entire sequence of *Shellder* with very few mismatches (see below). We also show results for *Spoink* and the *P*-element.

Why is *Shellder* absent in the assemblies but present in the short read data of the same strains? To answer this we first investigated how the assemblies were generated. In the paper where the assemblies were published (<https://genome.cshlp.org/content/33/4/587.short>) the authors state in the supplement: “TEFLoN first creates a pseudo-reference genome with TE sequences removed, then maps paired reads to the pseudo-reference genome and known TE sequences.”

Clearly, this assembly approach with TEFLoN will not identify novel TE sequences (or any novel sequences not present in the reference genome, for that matter). The absence of *Shellder* in the assembly of *dsim_28_F0* and *dsim_29_F0* is thus likely a technical artifact of the reference-guided assembly approach used for generating these two assemblies.

This issue is thus an interesting example, highlighting the dangers of relying on assemblies generated by reference-guided approaches.

Furthermore, I also failed to find intact copies in the *Drosophila cardini* genome (GCA_018903735.1_ASM1890373v1), in which authors found elements of a high similarity to the melanogaster *Shellder* (supplementary figure 12).

We do find two insertions of *Shellder* in *D. cardini* (GCA_018903735.1). Following the coordinates:

JAEIGM010000031.1:59782-66433

JAEIGM010000063.1:198948-205615

With a BLASTn search of these insertions against the *Shellder* consensus sequence, we find a query cover of 100% for both and a sequence similarity of 99.31% and 99.26%, respectively. The coordinates (bed file) and the sequences (FASTA file) of these two matches are provided at GitHub:

(<https://github.com/rpianezza/DoubleTrouble/tree/main/1-REVISION/Shellder-insertions>).

Overall, please clarify, wherever relevant, the exact region of TEs that was used to measure similarities between elements, and whether the insertions carry at least uninterrupted ORFs.

In the revised manuscript we provide a novel supplementary file (Supplementary file S3). This table provides for each of the three analyzed TEs (*Spink*, *Shellder*, *P*-element) and for each analyzed drosophilid assembly, the location of the best hit (i.e. the genomic coordinates), the sequence similarity, the length proportion (e.g. full-length or truncated), the Smith Waterman score and the similarity (shown in Fig 4A). This table thus contains the raw data for Fig 4A.

We also made it available here:

<https://github.com/rpianezza/DoubleTrouble/blob/main/1-REVISION/fig4A-rawdata.tsv>

Without those data available, I cannot evaluate the findings made in the paper.

minor comments

1. Please make available the multiple sequence alignments of the RT domains used to construct the phylogenetic trees.

The tree in Supplementary Fig. 1 is based on the RT domains. We made the msa available here:

https://github.com/rpianezza/DoubleTrouble/blob/main/trees/TEfamily-tree/protein-RT/D_mel_TE_RT_protein.MSA.

For the trees in Fig 4B and 4C we did not use the RT domains but the entire nucleotide sequence of the TEs. We made these msa files available here (nexus, xml):

<https://github.com/rpianezza/DoubleTrouble/tree/main/1-REVISION/TE-trees-files>

2. I cannot fully interpret the similarity plots.

What does similarity of <0.1 in a genome mean? Does that mean the most similar element in the genome has only such a little similarity to the TE in question? It is unlikely given that gypsy elements are highly abundant in most *Drosophila* genomes.

The “similarity” shown on the y-axis of fig. 4A is based on the Smith Waterman-score provided by RepeatMasker. Specifically, for each TE we take the best RM hit in a given

assembly (i.e. the sequence most similar to the TE found in that assembly) and the best RM hit across all assemblies. The similarity is the proportion between the best hit in a given assembly and the best hit in all assemblies (i.e. $\text{similarity} = \frac{\text{best_given_assembly}}{\text{best_all_assemblies}}$).

The similarity will be a value between 0 and 1, where 0 indicates that an assembly does not contain any subsequence matching with a given TE and 1 that an assembly has a perfect match with the given TE (provided the consensus sequence is present in at least one of the analyzed assemblies, which is the case in our scenario since we included the *D. melanogaster* strain from which the consensus sequence used for RM was generated).

The main advantage is that our approach allows summarizing the similarity between a TE and the best in an assembly with **a single descriptive number**. More naive approaches will at least require two numbers, like % of identity and length of the matching sequence, since for example a very short subsequence of a TE may have a perfect match with any assembly. With such an approach it is necessary to provide the sequence similarity **and** the length of the matching sequence. By contrast we think that our similarity score is an elegant approach, that allows us to capture the sequence similarity and the length of the match with a single number. To make it more clear that our similarity is based on the SW score rather than the actual sequence similarity we relabeled the y-axis in the revised manuscript to: “similarity (SW score)”.

Additionally, in the novel supplementary file (supplementary file S3) we provide for all matches shown in Fig 4A, the raw sequence similarity, the length of the match, the Smith-Waterman score and our similarity score. This allows readers to get an idea how our similarity score summarizes the sequence similarity and the length of the match. We also made the data available here:

<https://github.com/rpianezza/DoubleTrouble/tree/main/1-REVISION/TE-trees-files>

3. RepeatMasker uses many TE sequences that are similar to each other. This means that the lack of hits from a particular TE does not always mean that the genome has no similar elements. The examined genome may have hits from a very similar but different TE.

Please consider directly running blastn/tblastn of the TEs in question to rule out this possibility.

The reviewer is correct. During work for a previous paper, we also noticed that RepeatMasker solely provides the best match at a given genomic region. A good match with a TE (say Tirant) may not be reported if another TE (say *gypsy*) has a better match at the same coordinates.

For this reason we did not use the RM standard library, but rather a customized library with only the three TEs of interest (*Shellder*, *Spoink*, *P-element*) which have no relevant sequence similarity between each other.

Reviewer #3 (Remarks to the Author):

In the manuscript entitled “Double trouble: two retrotransposons triggered a cascade of horizontal transfers in *Drosophila* species within the last 50 years”, the authors demonstrated the cascading invasions of two retrotransposons, Shellder and Spink, among the worldwide *Drosophila* subspecies. To this end, they mined hundreds of short-reads and long-reads sequencing data from the time-series and geographic diverse *Drosophila* specimens. Their results reveal that the spread of these two retrotransposons was likely facilitated by human-mediated habitat expansion. Overall, this research is supported by extensive data from long-read assemblies and historical strains, providing a robust timeline for the invasion events.

1. In Figure 1, it would enhance the understanding of the invasion events if the authors could provide the chromosome ideograms displaying the locations of either full-length or fragment TEs overtime.

Figure 1 is based on short-read data. As a major technical limitation, short-read data do not permit distinguishing whether an insertion at a given site is due to a full-length or fragmented insertion. Solely long-read assemblies enable doing this. In supplementary table 1,5 and 7 we provide the positions of Spink and Shellder insertions in some assemblies.

As suggested by the reviewer we made some ideograms for a few long-read assemblies in *D. sim*. In this plot full-length and fragmented copies can be distinguished on the y-axis (full length $y=1.0$; fragmented $y<1.0$). However, we are not convinced that these figures are helpful to readers, for two reasons: First, the TE insertions are segregating (as expected for novel insertions) and thus at very different positions in each strain. Furthermore the assemblies have different numbers of contigs, and the contigs have different lengths, making it difficult to compare the locations. Therefore we opted to not include these figures in the manuscript.

D. sim SZ232

D. sim SZ129

2. The co-occurrence of Shellder and Spoink in certain species is interesting. It would be beneficial if the authors could provide a more definitive explanation.

We suggest that the co-occurrence of *Shellder* and *Spoink* in *D. mauritiana* and *D. sechellia* derive from an hybridization with *D. simulans*, which harbor both the TEs. However, this would not explain their co-occurrence in *D. simulans*, for which four possible hypothesis exist:

- 1) the two TEs transpose as a unit
- 2) the two TEs depend on each other, such that the products of one TE are required by the other TE

3) the two TEs were transmitted in a single horizontal transfer event from a Neotropical drosophilid (we added this hypothesis in the revised manuscript)

4) the two TEs invaded *D. simulans* around the same time by chance

We can rule out 1) because the insertion sites are different

We can rule out 2) because *Spink* but not *Shellder* is found in *D. melanogaster* and *Shellder* but not *Spink* in *D. teissieri*. This does not rule out that the products of one TE can support the other TE.

We did not find evidence for 3). Instead our data suggest that *Spink* and *Shellder* are derived from different Neotropical species (*Spink* from a species of the willistoni group and *Shellder* from a species of the cardini group). We can however not entirely rule out that an hitherto unsampled species acted as a donor of both TEs.

Which leaves 4) as the most likely explanation.

We however do not have definite proof for 4) and we thus prefer not to over interpret the results. In the revised manuscript we are discussing arguments in favor of and against these hypotheses in greater detail.

REVIEWER COMMENTS

Reviewer #1 (Remarks to the Author):

The authors addressed all my concerns and changes were made to the manuscript accordingly. I'm entirely satisfied.

Only 3 minor changes remain:

Lines 325 and 331:

Please define the abbreviation 'SW' in the manuscript or replace it with 'Smith-Waterman'.

We replaced 'SW' in the text with 'Smith-Waterman'

Line 521:

replace "repletagroup" with "repleta group"

Fixed

Line 524:

replace "than" with "then"

Fixed

Reviewer #2 (Remarks to the Author):

I thank the authors for addressing my questions point by point, with additional analyses and figures included.

I also thank that the authors made the raw data available in the git.

The authors addressed most of my questions. My major remaining question is the time scale of the horizontal transfer events.

That shellder/spoink are only found in recently collected melanogaster group strains, but not in earlier collections, suggests a recent introduction of these elements.

On the other hand, many of the insertions that the authors analysed and found most homologous to the consensus sequences in their respective genome assemblies do not code fully intact open reading frames. This means that despite the fact that there are multiple copies, none are currently active in those genomes.

This includes Shellder in Dsim sz232 and Shellder in D.cardini (GCA_018903735.1).

I thank authors for making the genomic sequences of the top hits available in the git. Although they contain several long ORFs, none of them appeared to have continuous POL ORFs.

I find it remarkable that the horizontal transfer events, which are presumably rare, introduced these elements into multiple different species, each of which led to expansion in their respective genomes, and then to demise and a partial degradation within a span of 50 years. We don't encounter the same extent of changes in transposon insertions in the Drosophila genome in laboratory settings.

Because of this extraordinary sequence of events, I suggest authors explaining the full account of these events in the manuscript including the demise of these elements. Especially, the current manuscript lacks information of whether the elements are currently transposing (namely the presence of fully intact copies) in which genomes.

We investigated the ORFs of *Shellder* in the two species.

We used the sequences of *Shellder* found in *D. sim* sz232 and used ORFfinder to identify ORFs. The screenshot below refers to the insertion JBBODM010000291.1:432122-438753 where we identified multiple long ORFs.

ORFs found: 139 Genetic code: 1 Start codon: stop-to-stop

ORF117 (646 aa) Display ORF as... Mark

```

>lcl|ORF117
FRVKKFRIVIDYRKLNEFTVDDRFPVPLNLSLDKLRGSOYFTLDDLAKG
FHQLVREEDRPKTFAPSTPSGHYEFVRMPFGLKNAPSTFQRLMNEVLKDH
INDNCVYVMDLILFSTLSQHMILLRKFRTLKEANLKIQVKKDFLKK
ETQFLGHLLTTHGKPNSSDKWNLQNLKLPKTAQILKSLGHTGFRYKRVF
RDYAKIAPMSRYLKNNETINTDDPSYIAAEKELKTLVNSPVLRYPNFS
KXFVTITDASNFVAGVLSQESHPIAYASRTLNQHECNYSYIEKELLAIV
WAKYFRPYYVYREFLESDDHPLKMLKMYTGKDISPRLGRWLNLSGY
NFHYEIKGNWNIADFLSRKEDEINLMEAESDEEDNKSLTETVHSNE
EDGFMMSILETAVRFKTIQIFTEKKPNTMVOVGRKRIYISKDKLENN
QAWLLRREITAGKIGVFSHLSQHEFYQFKLILTKYTSNPKVRFVKCTR
FYDIESEDELHTQIALFRNKSCHGCVATYQKLLKLYHPLKTHIR
  
```

Marked set (0)

SmartBLAST SmartBLAST best hit titles... | BLAST

BLAST Database: UniProtKB/Swiss-Prot (swissprot)

Label	Strand	Frame	Start	Stop	Length (nt aa)
ORF117	-	2	4251	2311	1941 646
ORF90	-	1	6133	5090	1044 347
ORF120	-	2	1356	427	930 309
ORF91	-	1	4999	4211	789 262
ORF99	-	1	2398	1946	453 150
ORF136	-	3	1781	1344	438 145
ORF126	-	3	5366	4992	375 124
ORF86	+	3	5664	6026	363 120
ORF84	+	3	5175	5453	279 92
ORF12	+	1	2533	2805	273 90
ORF43	+	2	2378	2635	258 85

In particular, the amino acid sequence of ORF117, highlighted in the screenshot, shows significant sequence similarity to a pol protein of *D. simulans* when aligned using BLASTp. Moreover, all three functional domains of the pol protein—reverse transcriptase, RNase H, and integrase—were detected in the BLASTp analysis.

Conserved domains on [lcl|ORF117]

View Standard Results ?

Local query sequence

In *D. cardini* we found a *Shellder* insertion that preserves the reverse transcriptase, the RNase H but not the integrase domain (JAEIGM01000031.1:59782-66433). This could however reflect that another species is the actual donor of *Shellder* (in the manuscript we argue that it could be a species of the *cardini*, *willistoni*, *repleta* group) or that the sequenced specimen of *D. cardini* just does not have a functional copy (but other, hitherto not yet sequenced, specimen of *D. cardini* might have one).

The presence of full Pol ORF in *D. simulans* suggests that *Shellder* could still be active in *D. simulans*. We do not find evidence for degradation of *Shellder* to such an extent that no functional copies remain in *D. simulans*. However, testing ongoing activity would involve scanning the offspring of crosses among strains having the TE (paternal) and strains not having the TE (maternal) for novel TE insertions (e.g. by massive sequencing). We argue that this is beyond the scope of the current manuscript where we solely aim to show that *Spoink* and *Shellder* recently spread in multiple species of the *D. melanogaster* group.

To address the concern, we added a sentence to the discussion: “Another interesting question is whether *Spoink* and *Shellder* are still active in the recently invaded species. Reciprocal crosses among strains with and without the TE could provide an answer.”.

Thank you for showing the mapping of short reads from dsim_28_F0 and dsim_29_F0 onto the transposon sequences.

I presume that the 88 Dsim genomes that were analysed in Supplementary Figure 4 are all short reads.

Please clarify this in the figure legend because in some cases, long-reads were used for Dsim strains.

The reviewer is correct, the strains in Supp. Fig. 4 are short-reads, we change the description of the figure to make it clear.

“Abundance of *Spoink*, *Shellder* and the *P*-element in 88 *D. simulans* (short-read data).”

The use of short reads brings a question of the sensitivity. The copy number estimation may be affected by the sequencing depth as well as the type of sequencing method (single read vs paired end, read lengths, etc). Please could you comment on this potential confounder?

The reviewer is correct in pointing out the challenges of working with short reads and repetitive elements.

Our pipeline for short-read quantification follows these steps to minimize those potential confounder effects:

1. **Trimming:** We trimmed all reads to the shortest length in the dataset, which in this case was 100 nt. Reads longer than this (e.g., 150 or 125 nt) were reduced to 100 nt, and smaller reads were discarded.
2. **Merging Paired-End Data:** For paired-end reads, we merged the data, though this process did not retain paired-end information, as the software DeviaTE treats all reads as single reads and does not utilize paired-end information.
3. **Mapping:** We used `bwa bwasw` to map the reads onto a file containing the consensus sequence of the TEs, as well as three single-copy genes. These single-copy genes were used for normalization. For instance, if single copy genes have a coverage of 10x and a TE has a coverage of 70x, then we infer there are approximately 7 copies of the TE. This normalization to the coverage of single-copy genes is an important step to reduce the impact of biases, as differences in the sequencing depth or methods will apply to both the reads aligning to TEs and SCGs.

A potential issue could arise if certain regions of the TE contain low-complexity sequences (e.g., poly-A stretches) or regions common to other TE families in the genome, which could artificially inflate copy number estimates. To avoid this, we generated plots like those shown in Supplemental Figure 3. In the case of *Shellder*, the transition from ~0 to ~10 copies and the absence of peaks suggest no homologous regions in the genome or low-complexity regions. For *Spoink*, although there are some peaks (lighter gray), the software distinguishes between high-quality (HQ) and low-quality mappings. We used only the HQ estimates, meaning our results were unaffected by these regions.

We believe our pipeline effectively addresses the reviewer's concerns. The significant increase from 0 to 10 copies supports our findings, despite any minor distortions in the copy number estimates.

We included the following in the Methods section to make the processing more clear: "We trimmed the short reads to a length of 100 nucleotides, discarding any shorter reads. Subsequently, we merged the paired-end files, as the following analyses did not require paired-end information."

Finally, the major findings of this work - e.g. the presence of *Spoink* and *Shellder* in some strains and their absence in others - is also supported by alternative approaches such as long-read assemblies (supplementary figs. 8, 10, 12).

Authors mentioned that they used only the three relevant TEs as baits for RepeatMasker searches.

Please state this more clearly in the method section

We followed reviewer's suggestion and we included the following sentence in the methods:
"We identified insertions of *Spink*, *Shellder* and *P*-element in these assemblies using RepeatMasker with a custom library consisting of the three TEs consensus sequences."

Reviewer #3 (Remarks to the Author):

The authors have addressed all my questions. I appreciate the detailed explanation regarding the co-occurrence of *Shellder* and *Spink* in the discussion. I also agree with the authors that the current ideogram data may not be particular helpful for the audience. I have no further questions.